## RESEARCH ARTICLE

# The cell-adhesion molecule Echinoid promotes tissue survival and separately restricts tissue overgrowth

Danielle C. Spitzer*, William Y. Sun, Anthony Rodríguez-Vargas and Iswar K. Hariharan‡

## ABSTRACT

The growth and survival of cells depends both on their intrinsic properties and interactions with their neighbors. In a screen of genes encoding cell-surface proteins for knockdowns that affect clone size or shape in mosaic *Drosophila* imaginal discs, we found that clones with reduced *echinoid* (*ed*) function are fewer and smaller, and are frequently eliminated during development. This elimination results, in significant part, from increased levels of apoptosis due to decreased Diap1 protein. We found that Hippo pathway activity is not decreased in *ed* mutant cells, as previously claimed, but is decreased in some of their immediate wild-type neighbors, consistent with the observed elimination of *ed* clones by a mechanism resembling cell competition. In contrast to the underrepresentation of *ed* clones, discs or compartments composed of mostly *ed* mutant tissue overgrow, despite having increased levels of apoptosis. The overgrowth results from a failure to arrest growth at the appropriate final size during an extended larval stage. Thus, *ed* has two distinct functions: an anti-apoptotic function via maintenance of Diap1 levels, and a function to arrest growth at the appropriate final size.

KEY WORDS: Echinoid, Adhesion, Growth, Hippo, Cell competition

## INTRODUCTION

Nowhere is the influence of neighbors more important than for cells in developing epithelia. In unicellular organisms, nutrient availability is the primary regulator of cell proliferation. In addition to nutrient availability, the survival, growth and proliferation of epithelial cells in multicellular organisms is regulated by various signals emanating from other cells (Lloyd, 2013). Such signals include diffusible factors that are secreted locally or circulate systemically (Conlon and Raff, 1999). The packing of cells into tissue layers generates mechanical forces that can influence cell proliferation and survival (LeGoff and Lecuit, 2016; Matamoro-Vidal and Levayer, 2019). Additionally, especially in epithelia, contacts between cells and their immediate neighbors, likely mediated by cell adhesion molecules, can determine whether cells remain within the layer or become extruded; extrusion often accompanies or promotes cell death (Bielmeier et al., 2016; Grata and Levayer, 2025).

The importance of the immediate neighbors of a cell in regulating its survival and proliferation is best illustrated by the phenomenon of cell competition, where 'loser' cells that are less fit for various reasons are eliminated only in the presence of faster-growing 'winner' cells (i.e. in genetic mosaics) (Morata and Ripoll, 1975) (reviewed by Amoyel and Bach, 2014; Baker, 2020; Cumming and Levayer, 2023; Nagata and Igaki, 2018). In their original discovery of this phenomenon, Morata and Ripoll (1975) found that cells heterozygous for a class of mutations known as *Minute*, which mostly disrupt ribosomal proteins (Marygold et al., 2007), are eliminated in the presence of wild-type cells. However, as a homogeneous population, they can generate animals of relatively normal size and shape, albeit much more slowly. Cells with *Minute* mutations upregulate a heterodimeric transcription factor composed of the Xrp1 and Irbp18 proteins, and this transcription factor is necessary for the elimination of *Minute/+* clones as well as for the growth alterations observed in *Minute/+* imaginal discs (Baillon et al., 2018; Langton et al., 2021; Lee et al., 2016, 2018). Clones of cells with mutations in genes encoding the E3 ubiquitin ligase component Mahjong (Ly et al., 2019; Tamori et al., 2010) or its binding partners are also eliminated via this pathway (Kumar and Baker, 2022).

Additionally, small patches of wild-type cells were shown to be eliminated when surrounded by cells having even modest increases in the level of Myc protein (de la Cova et al., 2004; Moreno and Basler, 2004). Cells capable of eliminating wild-type cells were dubbed 'supercompetitors'. Cells with mutations in the Hippo pathway that have increased activity of the transcriptional co-activator Yorkie (Yki) (Neto-Silva et al., 2010; Tyler et al., 2007), cells with increased Wnt signaling (Vincent et al., 2011), cells with increased Jak/Stat signaling (Rodrigues et al., 2012) and cells with reduced *crumbs* function (Hafezi et al., 2012) also behave as supercompetitors. Collectively, these findings show that the designation of 'winners' and 'losers' is not determined by the genotype of the cell itself but by how it compares to its neighbors. How this comparison is achieved is still not well understood since many mutations that alter competitive ability affect intracellular proteins and comparison presumably must occur at the cell surface. Once the comparison has happened, 'loser' cells upregulate specific isoforms of the cell-surface protein Flower (Rhiner et al., 2010). In addition, 'winner' cells express higher levels of the Toll ligand Spätzle, which might promote the loser cell fate in adjacent cells (Alpar et al., 2018).

Several phenomena resemble classical cell competition, some of which appear to be mechanistically distinct. Clones of cells with mutations in genes encoding proteins that regulate apicobasal polarity, such as *scribble* (*scrib*) and *discs large* (*dlg*), are eliminated during development (Brumby and Richardson, 2003). However, imaginal discs or compartments of discs that are entirely composed of mutant cells overgrow and the epithelia are often multilayered (Bilder et al., 2000). The elimination of *scrib* or *dlg* clones requires the TNF ortholog Eiger (Igaki et al., 2009). The loss of apicobasal polarity re-localizes

Department of Molecular and Cell Biology, University of California, Berkeley, CA 94720-3200, USA.
*Present address: Department of Biological Sciences, University of Pittsburgh, Pittsburgh, PA 15260, USA.

‡Author for correspondence (ikh@berkeley.edu)

D.C.S., 0000-0003-4827-1857; W.Y.S., 0009-0007-2008-5055; A.R.-V., 0000-0003-0306-2015; I.K.H., 0000-0001-6505-0744

**DEVELOPMENT**

the Eiger receptor Grindelwald, rendering it accessible to Eiger with the result that cells undergo apoptosis (de Vreede et al., 2022). Furthermore, a signaling event at the clone interface involving the ligand Stranded at second (Sas) and the receptor PTP10D, which promotes the elimination of polarity-deficient cells, has been proposed (Yamamoto et al., 2017); however, others have questioned a requirement for PTP10D (Gerlach et al., 2022). Mis-specified cells can also be eliminated by a JNK-dependent pathway (Adachi-Yamada and O'Connor, 2002) that involves the transmembrane protein Fish-lips (Adachi-Yamada et al., 2005). Studies of this 'interface surveillance' phenomenon indicate that differences in levels of cell-surface molecules can induce cell elimination (Fischer et al., 2024; Prasad et al., 2023). There has also been an increasing appreciation of the importance of mechanical forces in eliminating slowly proliferating cells (Levayer et al., 2016; Mao et al., 2013; Marinari et al., 2012; Shraiman, 2005), but these mechanisms on their own cannot easily account for observations that most, if not all, cells of specific genotypes are selectively eliminated.

One class of proteins that likely plays an important role in mediating heterotypic interactions at clone interfaces are cell-surface proteins, particularly cell-adhesion molecules. The *Drosophila* genome encodes over 100 proteins containing cadherin motifs or Ig-loops, which are commonly used for cell-cell adhesion (Hynes and Zhao, 2000; Vogel et al., 2003). Although many have been studied extensively in neuronal contexts, relatively few have been examined for a role in epithelial cell survival or proliferation (Finegan and Bergstrahl, 2020; Fischer et al., 2024), and fewer yet in a mosaic context. Here, we describe a genetic screen where we reduced the function of individual cell-surface proteins in clones of cells in the wing imaginal disc and identified those that affect clone size or shape. Of these, we focus on Echinoid (Ed) because its depletion in clones results in clone elimination, while depletion in the entire disc results in overgrowth. We demonstrate a role for Ed in maintaining levels of the anti-apoptotic protein Diap1 in cells and also a separate role in arresting growth when a tissue reaches its final size.

## RESULTS

The goal of our screen was to identify cell-adhesion molecules that regulate the survival, proliferation or arrangement of epithelial cells, especially when those mutant cells are adjacent to wild-type cells. To that end, we generated clonal patches of cells with reduced levels of individual adhesion molecules amid wild-type cells in wing imaginal discs, and assessed the number, size and shapes of the mutant clones. We used the FLP-out Gal4 system (Pignoni and Zipursky, 1997) to activate Gal4 expression in clones of cells, which drove expression of an RNAi transgene under the control of Gal4-responsive UAS elements (Brand and Perrimon, 1993; Fischer et al., 1988). The cells expressing the RNAi transgene were marked by expression of a fluorescent protein (UAS-GFP or UAS-RFP). The size and shape of clones were compared to those expressing an RNAi transgene directed against the *white* (*w*) gene (UAS-w-RNAi).

For our screen, we initially compiled a list of 153 genes that encoded known or putative cell-adhesion molecules (Fig. 1A; Table S1). This candidate list included genes with cadherin or immunoglobulin domains identified by Hynes and Zhao (2000) or Vogel et al. (2003), as well as others that we included based on reports in the literature. Since we conducted our screen in the wing imaginal disc, we excluded 43 genes that were not expressed in wing discs using the single-cell RNAseq data of Everetts et al. (2021) and 10 genes whose localization and function precluded a role in cell-cell adhesion (e.g. sarcomere components) (Table S2). Of the remaining 100 genes, we screened 74 genes using 90 different RNAi lines (Table S3).

While expression of most RNAi lines did not cause an obvious change in the number, size or shape of clones when compared to the expression of w-RNAi (Fig. 1B), five lines caused a reduction in clone size, two affected clone shape without reducing size, and two reduced clone size and generated rounder clones (Fig. 1A). Those that caused a marked reduction in clone number and size were *amalgam* (*ama*), *beaten path Vc* (*beat-Vc*), *sidestep VII* (*side-VII*), *Contactin* (*Cont*) and *shotgun* (*shg*) (Fig. 1C-G). Of these, only *ama* and *shg* were also found to impede cell growth or viability when knocked down in the entire posterior compartment rather than in clones, as assessed by an obvious decrease in posterior compartment size (Fig. S1), indicating that the clonal phenotype can differ from that elicited by more-widespread knockdown. Amalgam is a secreted protein whose best-characterized function is in axonal fasciculation (Frémion et al., 2000). Amalgam also has a role in regulating myoblast proliferation in leg discs and their interactions with tendon cells (Moucaud et al., 2024). Beat and Side family proteins interact with each other to facilitate cell adhesion. An analysis of physical interactions between Beat and Side proteins did not predict an interaction between Beat-Vc and Side-VII (Li et al., 2017). Contactin is a GPI-anchored protein needed to organize septate junctions of epithelial cells (Faivre-Sarrailh et al., 2004). *shg* encodes the *Drosophila* ortholog of E-cadherin, a homophilic adhesion molecule and a key component of adherens junctions (Oda et al., 1994; Tepass et al., 1996). Clones expressing RNAi constructs targeting *fat* (*ft*) (Fig. 1H) and *dachsous* (*ds*) (Fig. 1I) are rounder and larger than w-RNAi clones. Fat and Dachsous are atypical cadherins whose extracellular domains bind to each other; their role in regulating cell proliferation and cell division orientation has been studied extensively (reviewed by Fulford and McNeill, 2020; Thomas and Strutt, 2012). Knockdown of two genes, *echinoid* (*ed*) (Fig. 1J) and *off-track2* (*otk2*) (Fig. 1K,L) generated fewer clones that were typically smaller and rounder than w-RNAi clones (Fig. 1B). *otk2* clones were sometimes extruded as cysts (Fig. 1L,L'). The function of Otk2 is not well understood other than it likely functions as a co-receptor for the Wnt2 ligand (Linnemannstöns et al., 2014). Reduced growth of *ed* clones in imaginal discs has been noted previously (Escudero et al., 2003; Wei et al., 2005) but is not easily reconciled with some other functions of Ed (see below).

The Ed protein has an extracellular domain that includes seven Ig repeats and three FN-III repeats. Its 315 amino acid intracellular domain includes a C-terminal PDZ-binding protein that can interact either with Bazooka (Baz) or Canoe (Canoe) (Bai et al., 2001; Wei et al., 2005). Ed molecules on adjacent cells form homotypic adhesions via their extracellular domains (Rawlins et al., 2003b). *ed* clones are rounded with smooth outlines, in contrast to the irregular ('wiggly') boundaries of wild-type clones (Wei et al., 2005). *ed* mutant cells fail to assemble proper adherens junctions (AJs) at interfaces with wild-type cells (Chang et al., 2011; Laplante and Nilson, 2006; Wei et al., 2005). This differential adhesiveness likely causes these cells to remain together and sort away from wild-type cells (Steinberg, 1963; Townes and Holtfreter, 1955). Additionally, an actomyosin cable forms at Ed expression boundaries in the wild-type cells, potentially acting as a 'mechanical fence' (Chang et al., 2011; Laplante and Nilson, 2006, 2011; Lin et al., 2007; Wei et al., 2005). Together, these properties explain the roundness and apical smoothness of *ed* clones.

Ed is also thought to function as a signaling molecule. *ed* mutations were originally identified as dominant enhancers of a hypermorphic EGF receptor allele (Bai et al., 2001). Ed was subsequently shown to negatively regulate EGF receptor (EGFR) signaling (Bai et al., 2001; Fetting et al., 2009; Ho et al., 2010; Islam et al., 2003; Rawlins et al., 2003a; Spencer and Cagan, 2003) and positively regulate Notch signaling (Ahmed et al., 2003;

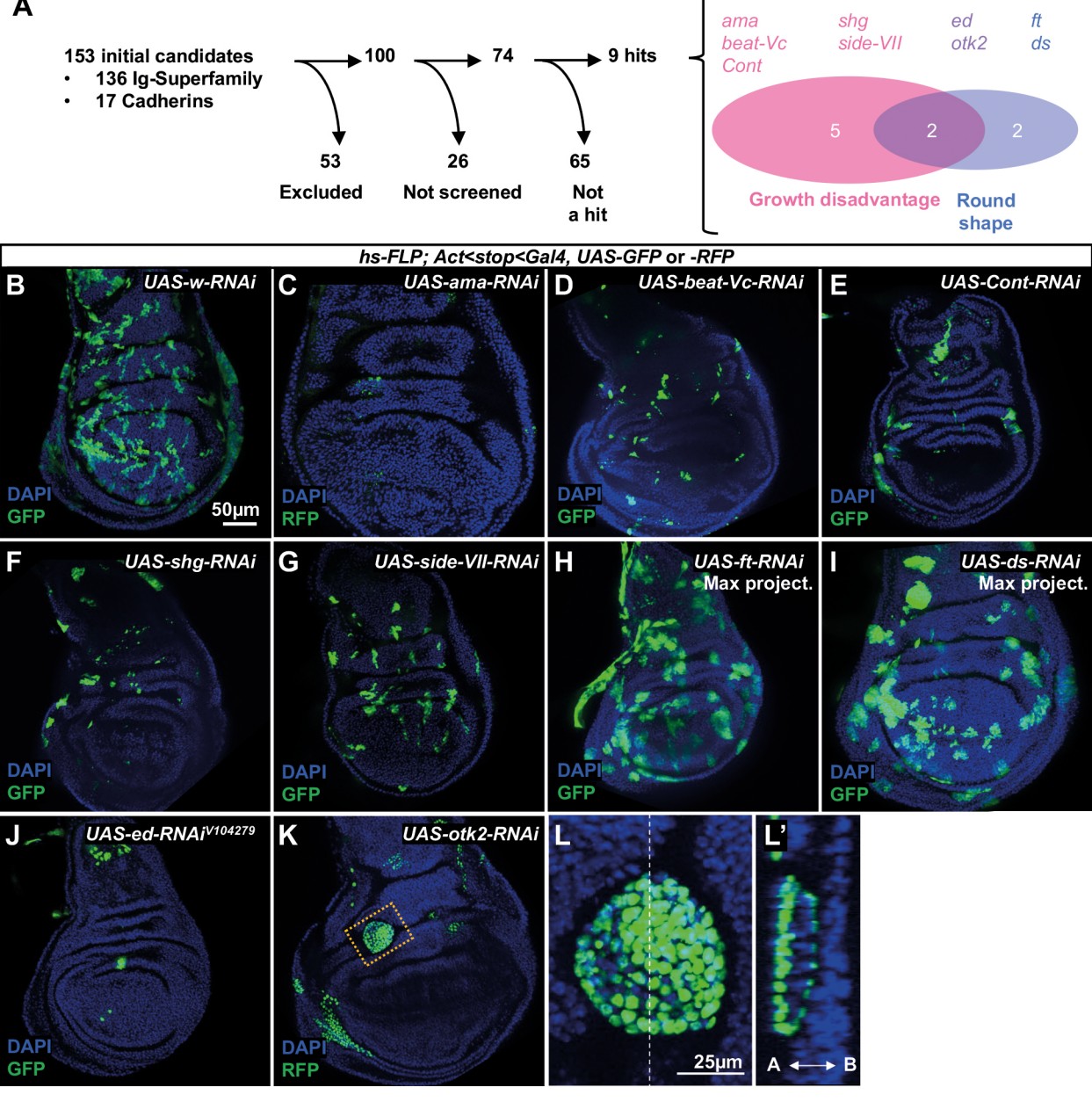

**Fig. 1. Clonal phenotypes observed in RNAi screen.** (A) Summary of the screen. (B-L′) Phenotypes of imaginal discs containing clones generated with a FLP-out Gal4 and *UAS-RNAi* transgenes. Clones are marked by GFP or RFP, as indicated. A cyst-like clone in K is shown at higher magnification in L and an orthogonal view is provided in L′. Scale bar in B applies to B-K. Scale bar in L applies to L,L′.

Escudero et al., 2003; Rawlins et al., 2003b), possibly by promoting endocytosis of EGFR and Delta, respectively. More recently, reduced *ed* function has been shown to cause tissue overgrowth by decreasing signaling via the Hippo pathway (Yue et al., 2012). Ed interacts physically with multiple Hippo-pathway components, promotes the stability of Salvador (Sav), and curtails the expression of Yki-target genes that normally promote growth and cell survival. Since both the EGFR pathway and Yki promote cell survival and proliferation, and their activity would be predicted to increase in *ed* mutant tissue (Bergmann et al., 1998; Díaz-Benjumea and García-Bellido, 1990; Huang et al., 2005; Kurada and White, 1998), the underrepresentation of *ed* mutant tissue in clones is not easily explained. We therefore decided to examine the properties of *ed* mutant tissue in greater detail.

**Clones of *echinoid* mutant cells in epithelia are eliminated by a process that resembles cell competition**

We generated clones using four different *ed-RNAi* lines (Fig. 2A-D) to evaluate the strength of knockdown and the resulting phenotype. Clones were induced 72 h after egg lay (AEL) and imaginal discs were dissected at 120 h AEL. Lines V104279 (Fig. 2A) and V3087 (Fig. 2B) generated fewer and smaller clones than controls (compare to Fig. 1B), with almost undetectable Ed protein (Fig. 2A′,A″,B′,B″). BL38243 generated clones that were not smaller and had wiggly outlines, much like wild-type clones (Fig. 2D); the knockdown was least effective in this line based on detectable anti-Ed antibody staining (Fig. 2D′,D″). V938 had an intermediate effect (Fig. 2C-C″). We also used mitotic recombination to generate *ed* clones with the MARCM method (Fig. 2E-H). We observed that homozygous mutant MARCM clones of two different null alleles of *ed* (*ed^IF20^* and

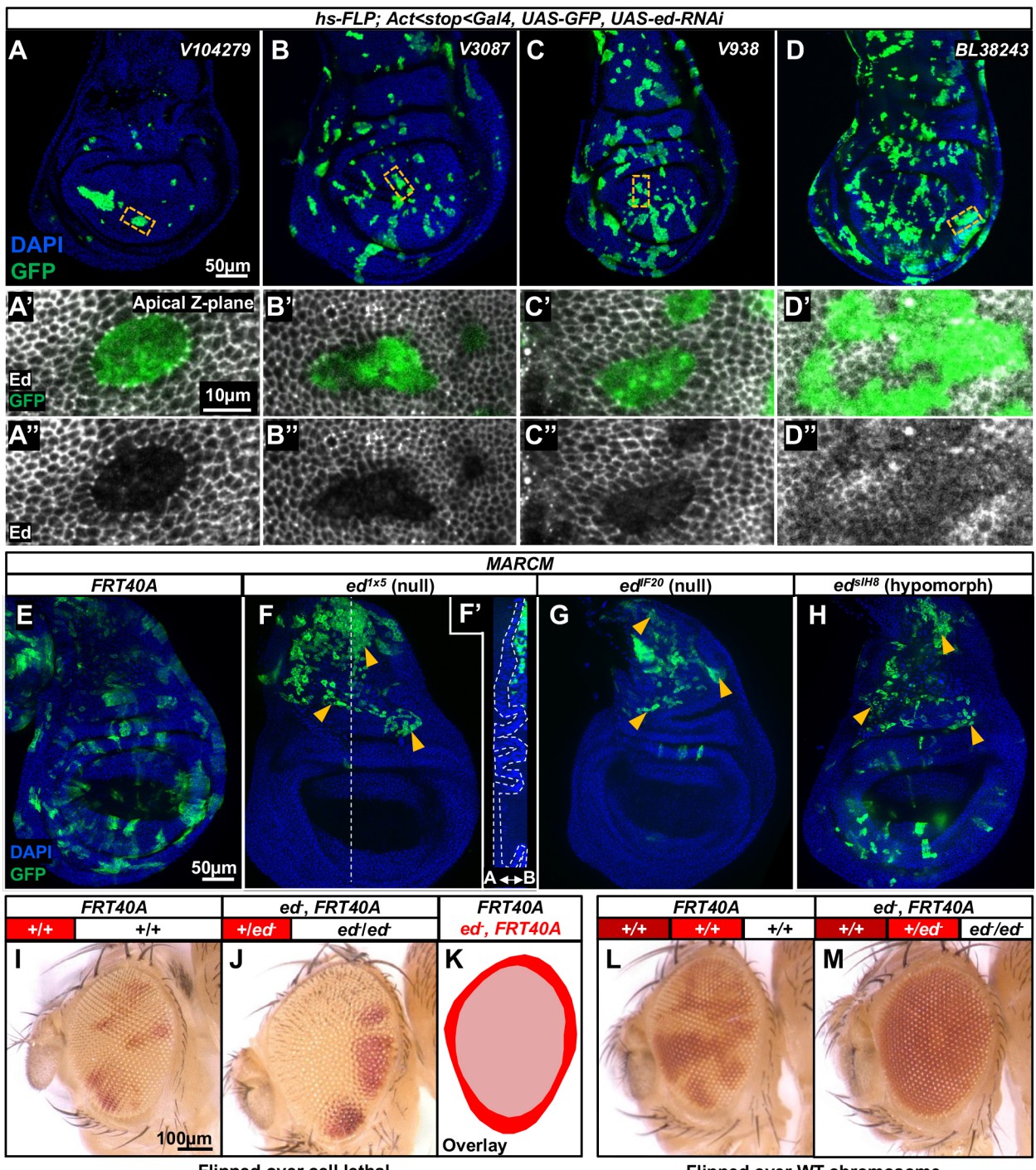

**Fig. 2. Clones of *echinoid* mutant cells are eliminated when surrounded by wild-type cells.** (A-D) Imaginal discs containing GFP-marked *ed-RNAi* clones. Four different *UAS-ed-RNAi* transgenes were used. (A′-D″) Clones stained with an anti-Ed antibody (outlined in A-D) at higher magnification. (E-H) Clones that are homozygous for the chromosome arm bearing *FRT40A* alone (E) or *FRT40A* and an allele of *ed* (F-H) generated using the MARCM method (Lee and Luo, 1999). Homozygous mutant clones are positively marked with GFP. Note the near absence of clones in the epithelium with the two null *ed* alleles (F,G) and the presence of myoblast clones underlying the notum (arrowheads in F-H). An orthogonal view is shown in F′ with the disc epithelium outlined; the GFP-marked myoblasts are located basal to the epithelium. The A/B double-headed arrow in F′ indicates apicobasal orientation of the wing disc proper epithelium. Both epithelial and myoblast clones are observed with the hypomorphic allele (H). (I-K) Clones homozygous for either a wild-type chromosome arm distal to *FTR40A* (I) or *ed^IF20, FRT40A* (J), marked white, generated using *eyFLP*. The tester chromosome carries a recessive cell lethal allele *l(2)cl-L3^1*, resulting in the absence of wild-type twin spots when homozygous. (K) An overlay of the overgrown eye containing *ed* clones and the normally sized eye containing wild-type clones. (L,M) Clones generated using *eyFLP* using a wild-type tester chromosome that does not carry a recessive cell lethal mutation. *FRT40A* clones (L) and *ed^IF20 FRT40A* clones (M) are white, while the wild-type twin spots appear red. Note the almost complete absence of homozygous *ed/ed* tissue in eyes that contain wild-type twin spots.

$ed^{1x5}$) (Bai et al., 2001; de Belle et al., 1993) were completely absent from the disc epithelium (Fig. 2F,G). $ed^{IF20}$ and $ed^{1x5}$ have early stop codons and are predicted to encode only small portions of the extracellular domain (Escudero et al., 2003). Clones that are homozygous for the hypomorphic allele $ed^{slH8}$ (de Belle et al., 1993), which has a missense mutation that changes a conserved cysteine residue in the sixth extracellular Ig domain to serine (Escudero et al., 2003), were observed, but were less frequent and smaller than wild-type clones (Fig. 2H). In all these experiments, large clones of myoblasts are often observed beneath the epithelium in the notum region of the disc (Fig. 2F'). Although Ed is expressed in wing disc-associated myoblasts based on single cell RNA-seq data (Everetts et al., 2021), and is required for targeting and morphogenesis of some embryonic and larval body wall muscles (Swan et al., 2006), $ed$-depleted myoblast clones do not have an obvious growth or survival disadvantage. As described previously (Chang et al., 2011; Laplante and Nilson, 2006, 2011; Lin et al., 2007; Wei et al., 2005), an apical actomyosin cable was observed in epithelial $ed/ed$ clones at the mutant clone periphery (Fig. S2).

Yue et al. (2012) previously reported that $ed/ed$ clones generated during eye development were overgrown. An important difference between that experiment and our experiments is that their clones were generated in a stock where the wild-type chromosome carried a recessive cell-lethal mutation. Thus, following $eyFLP$-driven mitotic recombination, the eye disc would contain mostly $ed/ed$ tissue as well as a small amount of $ed/+$ tissue that had not undergone mitotic recombination. Consistent with their observations, when the wild-type chromosome carries a recessive cell-lethal mutation, we also find that eyes composed almost entirely of $ed^{IF20}/ed^{IF20}$ tissue are overgrown (Fig. 2I-K). However, when a recessive cell-lethal mutation is not used to eliminate the wild-type twin clones, we observed almost no homozygous $ed/ed$ tissue (Fig. 2L,M). Thus, $ed/ed$ tissue can survive, and even overgrow, when it makes up most of the disc. In contrast, $ed/ed$ tissue is extremely underrepresented in the presence of wild-type tissue. This indicates the importance of wild-type cells in the elimination of $ed/ed$ cells.

To further explore this phenomenon, we generated $ed$-RNAi clones at variable densities by expressing $hs$-$FLP$ for 12 min, 15 min or 30 min (Fig. 3A-F). With the shortest heat shock, there were far fewer $ed$-RNAi clones than $w$-RNAi (control) clones (Fig. 3A,B). As clone density increased with longer heat shocks until the clones collectively accounted for most of the tissue in the disc, the representation of $ed$-RNAi tissue and $w$-RNAi tissue was similar (Fig. 3C-F). Thus, the loss of $ed$ mutant tissue is most likely to occur when those clones are surrounded by wild-type cells.

In their original description of cell competition, Morata and Ripoll (1975) observed that $Minute$ clones were eliminated when generated early in development but were still observable when generated much later in development. To determine if this was also the case with $ed/ed$ clones, we generated clones either 48 h or 96 h AEL using mitotic recombination, and dissected discs at 120 h AEL (Fig. 3G). Thus, the discs developed for 72 h or 24 h after clone generation, respectively (Fig. 3H-K). Marked wild-type clones were much larger 72 h after clone generation (Fig. 3J) than 24 h after clone generation (Fig. 3H). In stark contrast, $ed/ed$ clones were observed readily 24 h after clone generation (Fig. 3I) but were mostly absent 72 h after clone generation (Fig. 3K). At 72 h after clone generation, their wild-type sister clones were similar in size to wild-type clones generated at similar times (Fig. 3J,K). Thus, $ed/ed$ clones proliferate for a short time and are then eliminated. The propensity of $ed/ed$ tissue to be eliminated when surrounded by wild-type tissue, and to survive and grow when it accounts for most of the tissue resembles other instances of cell competition. However, unlike $Minute/+$ clones, expression of the $Xrp1$-$lacZ$ reporter is not elevated in $ed/ed$ clones (Fig. S3A-D), indicating that $ed/ed$ clone elimination and $Minute/+$ clone elimination are somewhat mechanistically distinct.

## Increased apoptosis and reduced levels of Diap1 are observed in echinoid mutant tissue

In many situations where clones of cells of a certain genotype (e.g. $Minute/+$) are eliminated by cell competition, their elimination occurs by caspase-mediated apoptosis, and their death can be rescued by expression of the baculovirus p35 protein (Martín et al., 2009), which inhibits effector caspases (Hay et al., 1994). We therefore generated clones expressing either $w$-RNAi or $ed$-RNAi in the presence or absence of $p35$ (Fig. 4A-D). While expression of $p35$ modestly increases the recovery of wild-type clones (Fig. 4A,B), there was a dramatic increase in the recovery of clones expressing $ed$-RNAi (Fig. 4C,D), indicating that effector caspase activity is necessary for $ed$ clone elimination. Apoptotic cells in $Drosophila$ express a cleaved version of Death Caspase-1 (Dcp-1) (Song et al., 1997) that is recognized by the anti-Dcp-1 antibody. In discs containing $w$-RNAi clones, occasional Dcp-1 staining was observed (Fig. 4E,E'). In contrast, punctate Dcp-1 staining was observed within $ed$-RNAi clones, especially in basal focal planes (Fig. 4F,F'). These puncta were often at the clone edge where it abuts wild-type tissue, suggesting that competition with neighbors may be involved. Taken together, these observations indicate that cells with reduced $ed$ function die by apoptosis at an increased rate; this contributes significantly to $ed$ clone elimination.

An important regulator of apoptosis is the IAP protein Diap1 (Hay et al., 1995), which inhibits caspase activation. When discs containing $ed$-RNAi clones were stained with an anti-Diap1 antibody, a reduction in Diap1 levels was observed within many clones (Fig. 4G,G'). If Diap1 reduction has a role in promoting the death of $ed$ mutant cells, then restoring Diap1 levels should reduce or prevent clone elimination. To examine this possibility, we expressed $UAS$-$diap1$ concurrently with $UAS$-$ed$-$RNAi$, which resulted in the recovery of many $UAS$-$ed$-$RNAi$ clones (Fig. 4H,H'). This suggests that the reduced Diap1 protein level in $ed$ mutant clones has a role in their elimination. The $ed$-RNAi clones that were rescued by $diap1$ overexpression still had smooth outlines, indicating that this property of the clones is separable from the propensity of clones to be eliminated.

Although consistent with increased apoptosis, our finding that Diap1 levels were reduced in $ed$ clones was surprising because a previous study reported decreased activity of the Hippo signaling pathway and increased expression of Yki target genes, including $diap1$, in $ed$ clones (Yue et al., 2012). This prompted us to further investigate Hippo pathway activity in and around $ed$ clones.

## Yki-target genes are altered in and around ed clones

Clones mutant for Hippo pathway components have increased levels of Diap1 protein (Tapon et al., 2002) and increased expression of $diap1$ transcriptional reporters (Wu et al., 2003). Using a transcriptional reporter of $diap1$ that contains eight copies of a Hippo-responsive element (HRE) (Wu et al., 2008), we observed decreased reporter expression in $ed$-RNAi clones and sometimes observed increased expression in cells immediately outside the clone (Fig. 5A,A'); we also observe this 'border effect' in many clones when staining with anti-Diap1 (Fig. 5B,B'). Additionally, two other Yki-responsive transcriptional reporters, $fj$-$lacZ$ (Brodsky and Steller, 1996) and $bantam$-$lacZ$ (Herranz et al., 2012) (a direct reporter of $bantam$ transcription, not the $bantam$ sponge that is inversely correlated to $ban$ levels), are expressed at lower levels in $ed$-RNAi clones and at higher levels in some neighboring cells (Fig. 5C,C',D,D').

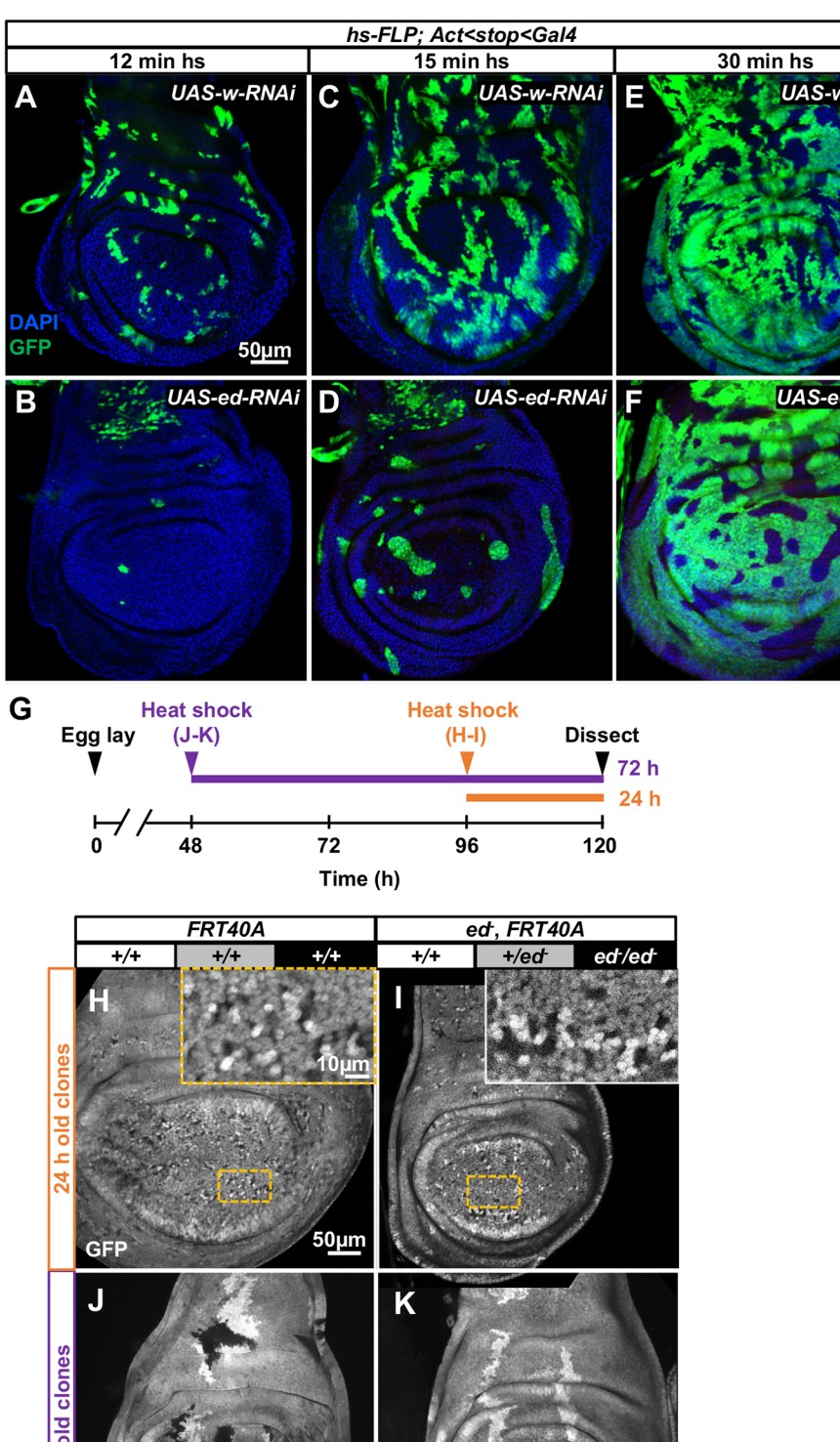

**Fig. 3.** ***echinoid* clones are generated and then die, especially when more wild-type tissue is present.** (A-F) Wing discs containing GFP-marked clones expressing either *UAS-w-RNAi* (A,C,E) or *UAS-ed-RNAi* (B,D,F). Heat shocks of 12 min (A,B), 15 min (C,D) and 30 min (E,F) generate clones at progressively higher density. (G-K) An experiment to examine *ed/ed* clones 24 h and 72 h after generation by mitotic recombination. The design of the experiment is shown in G. Discs were dissected 120 h AEL and clones were induced either 24 h (H,I) or 72 h (J,K) before dissection. Clones generated using the *FRT40A* chromosome (H,J) are compared to clones homozygous for *ed^{IF20}* (I,K). The outlined areas are shown in the insets. Scale bar in H applies to H-K. Scale bar in A applies to A-F.

Since nuclear Yki drives the Hippo pathway-mediated expression of *diap1*, *fj* and *bantam*, we examined Yki localization in and around *ed-RNAi* clones. Using a GFP-tagged Yki (Fletcher et al., 2018) and an anti-Yki antibody, we observed no difference in Yki localization between *ed-RNAi* clones and neighboring wild-type cells (Fig. 5E-E‴). This result contrasts with data presented by Yue et al. (2012), which showed strong nuclear localization of Yki in *ed* cells and diffuse staining in wild-type cells. We found their result surprising, given that Yki antibody stains typically produce a 'honeycomb-like' pattern in wild-type cells in the disc proper of the wing imaginal disc due to nuclear exclusion; nuclear relocalization of Yki caused by Hippo pathway mutations generally results in more

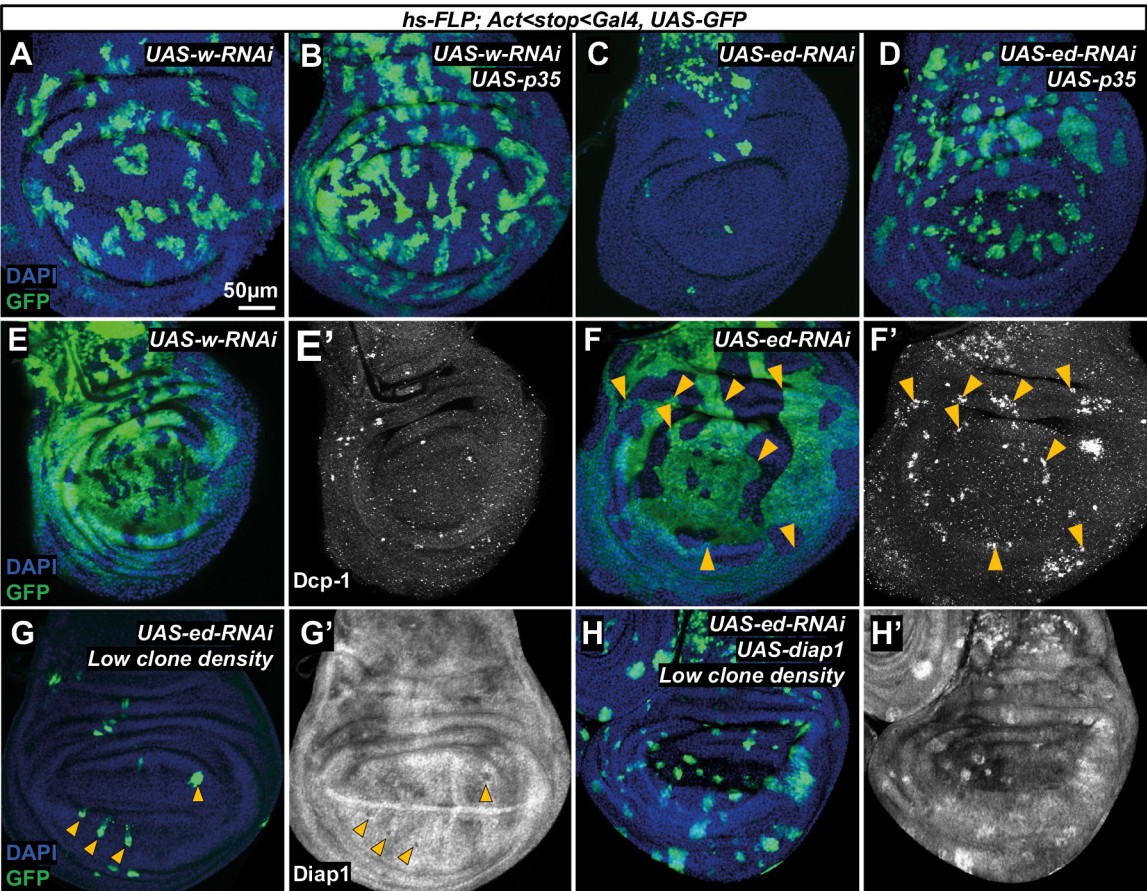

**Fig. 4. _echinoid_ mutant tissue has higher apoptosis levels and lower Diap1 levels.** (A-D) Wing discs containing GFP-marked clones that express either _UAS-w-RNAi_ (A,B) or _UAS-ed-RNAi_ (C,D). Clones that also express _UAS-p35_ are shown in B,D. (E-F′) Imaginal discs containing GFP-marked clones expressing _UAS-w-RNAi_ (E) or _UAS-ed-RNAi_ (high clone density) (F). Anti-Dcp-1 staining of the same discs are shown in E′,F′. Images were taken at a basal _z_-plane where Dcp-1 staining was most prominent. Arrowheads in F,F′ highlight examples of anti-Dcp1 staining near clone boundaries. (G-H′) Discs expressing UAS-ed-RNAi alone (low clone density) (G) or both _UAS-ed-RNAi_ and _UAS-diap1_ in GFP-marked clones, stained with anti-Diap1 (G′,H′). Arrowheads in G,G′ show the location of clones where reduced Diap1 is apparent. 'Rescued' clones (H,H′) still have smooth borders.

uniform staining of the cytoplasm and nucleus, rather than in strong, distinctly nuclear staining (see, for example, Fig. 3C,D in Dong et al., 2007). In summary, because we observed lower expression of at least three Yki-dependent reporter genes and no obvious change in Yki localization within _ed_ clones, our results are inconsistent with the previous assertion (Yue et al., 2012) that _ed_ mutant tissue has increased levels of nuclear Yki and increased expression of Yki-target genes.

Border effects like those observed in _ed_ clones have been described previously in response to manipulations of the Fat/Dachsous (Ft/Ds) pathway (Matakatsu and Blair, 2012; Willecke et al., 2008). To investigate a possible role for this pathway, we stained discs containing _ed-RNAi_ clones with anti-Ft. We observed increased staining in the clones (Fig. 5F,F′) that might be due to increased Ft levels, concentration of Ft protein into a smaller area of apically constricted membrane, or a combination of both. Apical constriction within _ed_ clones, as well as apical expansion of their immediate neighbors, has been noted previously (Chang et al., 2011; Laplante and Nilson, 2006; Wei et al., 2005). Apical expansion could dilute the Ft concentration in the cells immediately adjacent to the clone. A relative reduction in Ft levels in the border cells would be expected to reduce Hippo pathway activity (Bennett and Harvey, 2006; Cho et al., 2006; Silva et al., 2006; Willecke et al., 2006) and increase expression of Yki-target genes in those border cells, as we have observed, thus providing a

potential mechanism through which cells with reduced _ed_ function can interact with their wild-type neighbors. In the absence of direct evidence that the increased expression of Yki target genes in wild-type cells bordering the clone is caused by alterations in Ft distribution, a variety of other mechanisms, ranging from changes in various signaling pathways to alterations in cell adhesion, could apply.

We tested whether overexpression of _ed_ had opposite effects on Yki targets to those observed when _ed_ function was reduced. Diap1 protein levels were unaffected in _ed_-overexpressing clones (Fig. S4A,A′) but reduced when _ed_ was overexpressed in the entire posterior compartment (Fig. S4B,B′). In contrast, _ban-lacZ_ expression was increased in _ed_-overexpressing clones (Fig. S4C,C′). Expression _ex-lacZ_ was increased by _ed_ overexpression in the entire posterior compartment, particularly in the _ed_-overexpressing cells abutting wild-type cells (Fig. S4D,D′). These results with _ed_ overexpression also do not support a simple relationship between Ed and Yki where increased levels of Ed would reduce Yki activity, as Yue et al. (2012) have proposed.

### Imaginal discs with reduced _ed_ function grow more slowly but fail to arrest their growth at the appropriate size

Previous work has indicated that reducing _ed_ function in most, or all, cells of the imaginal disc results in overgrowth (Bai et al., 2001; Yue et al., 2012). This, at first glance, seems inconsistent with the

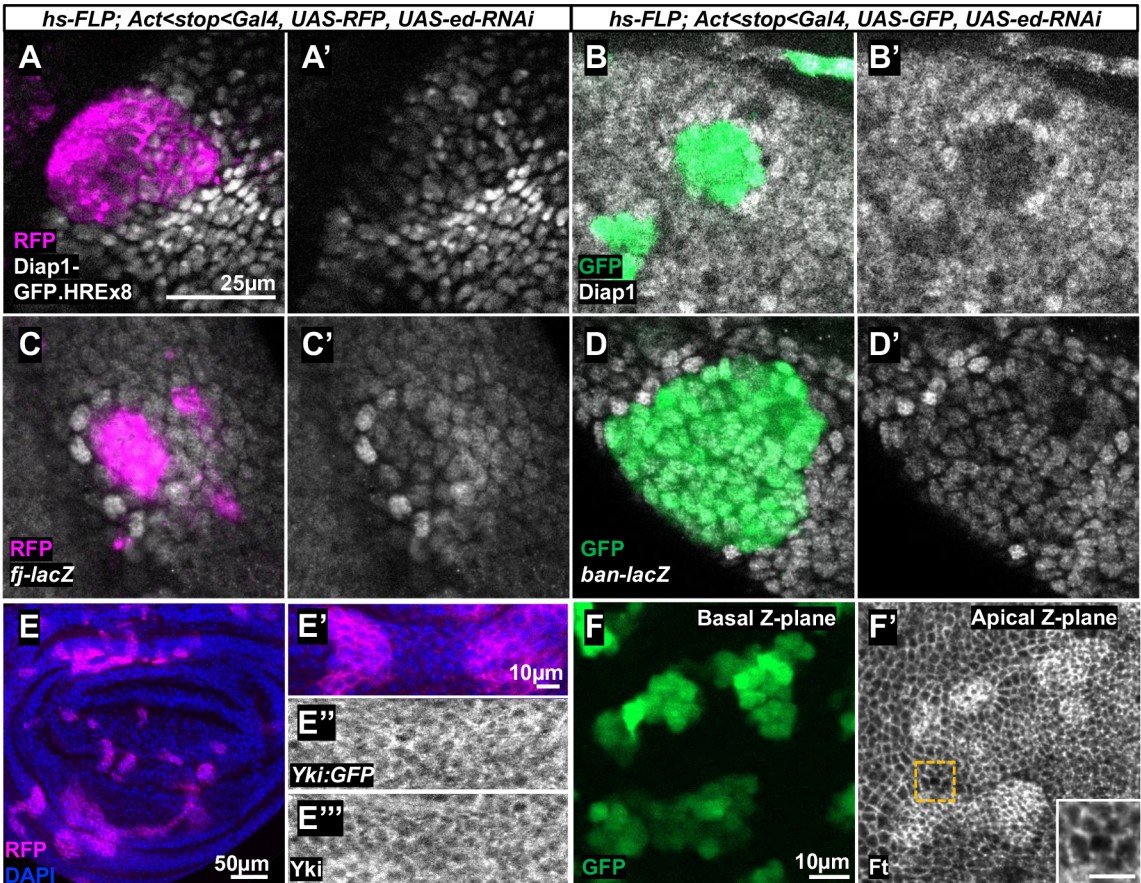

**Fig. 5. Hippo pathway reporters are altered in and around *echinoid* clones.** (A,A′) RFP-marked (A) *ed-RNAi* clone shows decreased expression of a *diap1* transcriptional reporter generated using eight copies of the Hippo-response element (HRE) from the *diap1* locus (A′). (B,B′) GFP-marked (B) *ed-RNAi* clones stained with anti-Diap1 (B′). (C,C′) RFP-labeled (C) *ed-RNAi* clone shows decreased *fj-lacZ* expression (C′). Wild-type cells adjacent to the clone show increased *fj-lacZ* expression. (D,D′) GFP-labeled (D) *ed-RNAi* clone shows decreased *ban-lacZ* expression (D′). Wild-type cells adjacent to the clone show increased *ban-lacZ* expression. (E-E‴) RFP-labeled (E,E′) *ed-RNAi* clones show no obvious alteration in the localization of either GFP-tagged Yki (E″) or anti-Yki staining (E‴). The region shown at higher magnification in E′-E‴ is outlined in E. (F,F′) GFP-marked (F) *ed-RNAi* clones stained with an anti-Ft antibody (F′). The inset shows a higher magnification of the boundary of the clone. F and F′ show different *z*-planes of the same image because the Ft signal is located at a *z*-plane with weak GFP signal. Scale bar in A applies to A-D′. Scale bar in F′ (inset) is 5 μm.

phenotype we observed in clones. *ed* clones in mosaic tissue have increased levels of cell death and reduced Diap1 levels. Does this also apply to entire *ed*-depleted compartments in the wing disc? We investigated this by expressing *ed-RNAi* using *hh-Gal4*, which is expressed in the entire posterior compartment (Fig. 6A-D). In these wing discs, we observed increased levels of apoptosis (Fig. 6A,B) and reduced Diap1 levels (Fig. 6C,D). To study the growth properties of these discs, we examined them at different stages of development (Fig. 6E-K). At 120 h AEL, when control larvae (*hh-Gal4, UAS-w-RNAi*) have reached the late third instar, the posterior compartments of discs of mutant larvae (*hh-Gal4, UAS-ed-RNAi*) were much smaller (Fig. 6F,H). However, the discs of mutant larvae continued to grow through an extended larval stage and could reach a size much larger than ever observed in wild-type discs (Fig. 6I-K). Interestingly, although the *ed-RNAi* is only expressed in the posterior compartment, increased growth was also observed in the anterior compartment. This observation is consistent with the phenomenon of positional accommodation that can occur in imaginal discs, where changes to growth in mosaic tissues exerts a non-autonomous effect on the growth of wild-type tissues (Díaz-Benjumea et al., 1989; García-Bellido, 2009; García-Bellido et al., 1994). This may be mediated by increased morphogen production from the enlarged posterior compartment. We observed a relatively

normal pattern of phospho-Mad staining in overgrown discs generated using *en-Gal4* (which has a similar expression pattern to *hh-Gal4*), indicating that Dpp production scales with the larger discs (Fig. S5).

Ilp8 protein is secreted by tissues undergoing repair after damage and by tissues that grow excessively due to genetic perturbations such as mutations that disrupt apicobasal polarity. Ilp8 production extends the larval phase of development by delaying the surge in ecdysone production that triggers entry into metamorphosis (Colombani et al., 2012; Garelli et al., 2012). Expression of *ed-RNAi* in the posterior compartment results in increased expression of *ilp8-GFP* (Fig. 6L,M), indicating a likely role for *ilp8* in extending the larval phase of development in *hh-Gal4, UAS-ed-RNAi* discs. Thus, at least under the conditions of this experiment, *ed* mutant tissue grows more slowly than wild-type tissue (likely due to the increased cell death). However, pupariation is delayed and discs can eventually grow to be unusually large, indicating a defect in the mechanism that normally arrests growth at the appropriate final size.

### Echinoid regulates the size and shape of the adult wing
If one role for *ed* is the arrest of growth at the correct final size, then reducing *ed* function would be expected to increase wing size.

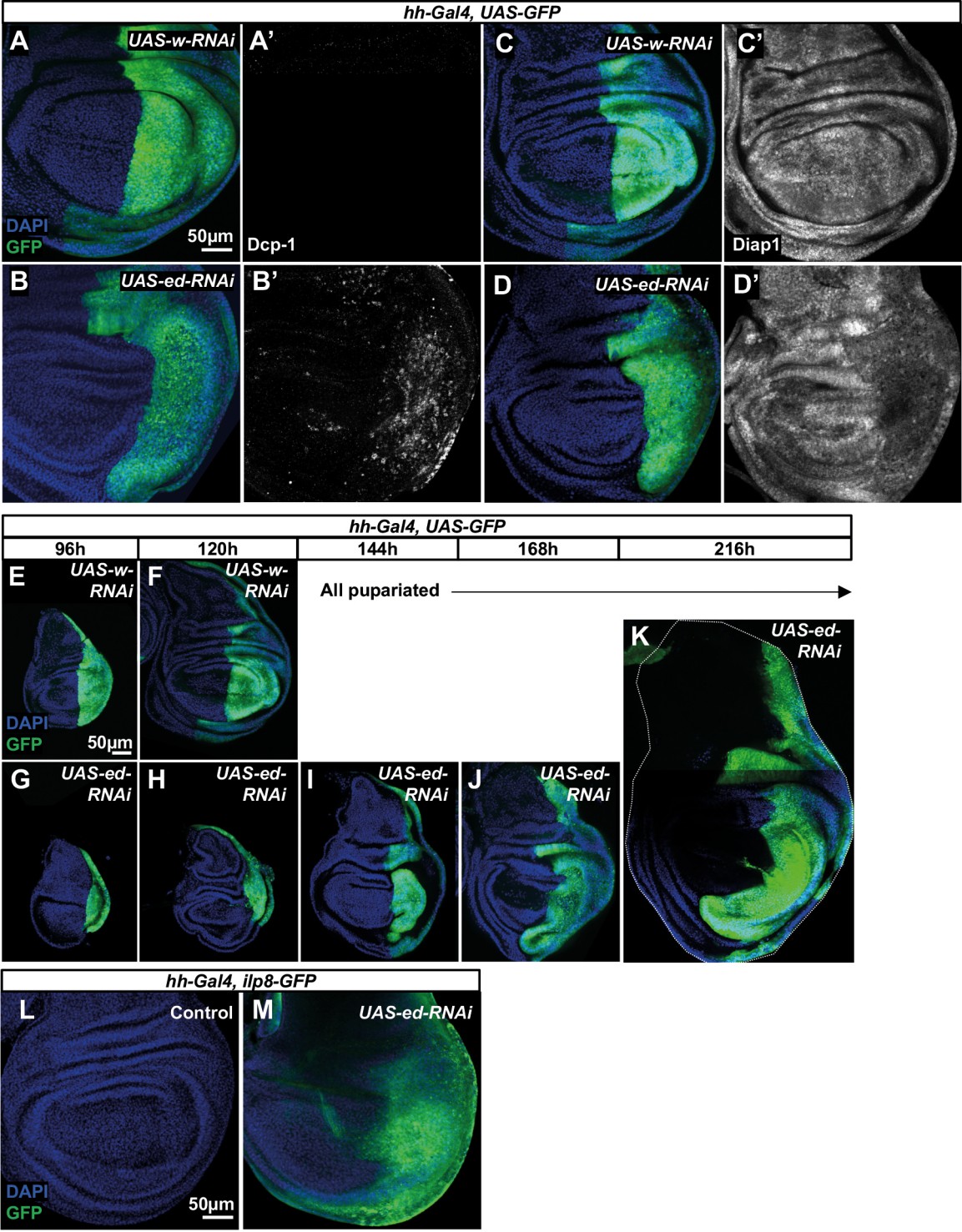

**Fig. 6. Characteristics of discs with compartment-wide *echinoid* loss.** (A-D′) Effect of reducing *ed* function in the posterior compartment on apoptosis and Diap1. (A-B′) *hh-Gal4* drives expression of *UAS-GFP* (A,A′) or *UAS-GFP* and *UAS-ed-RNAi* (B,B′). Apoptotic cells are visualized with anti-Dcp1 (A′,B′). (C-D′) Discs expressing *w-RNAi* (C) or *ed-RNAi* (D) in the entire posterior compartment stained with anti-Diap1 (C′,D′). Scale bar in A applies to A-D′. (E-K) Time course of growth of imaginal discs expressing either *w-RNAi* (E,F) or *ed-RNAi* (G-K) in the GFP-marked posterior compartment. All larvae expressing *w-RNAi* pupariated soon after 120 h. Much older larvae were observed in the population expressing *ed-RNAi*; examples of discs from these larvae are shown. Scale bar in E applies to E-K. (L,M) Effect of reducing *ed* function in the posterior compartment on Ilp8 expression. *hh-Gal4, Ilp8-GFP* (L) discs do not express detectable levels of Ilp8-GFP, but when *hh-Gal4* drives expression of *ed-RNAi* (M), Ilp8-GFP is elevated in a pattern that coincides with the expected location of the unmarked posterior compartment. Scale bar in L applies to L,M.

Adult flies homozygous for null alleles of *ed* were not obtained, implying lethality at an earlier stage of development. We therefore examined the size of adult wings in a null/hypomorph heteroallelic combination, *ed^{IF20}/ed^{sIH8}*. Wings of these flies were 9% larger by area than wings of wild-type (*Oregon-R*) flies (Fig. 7A-D,D′,I). We also reduced *ed* function in the wing pouch using *nub-Gal4* or

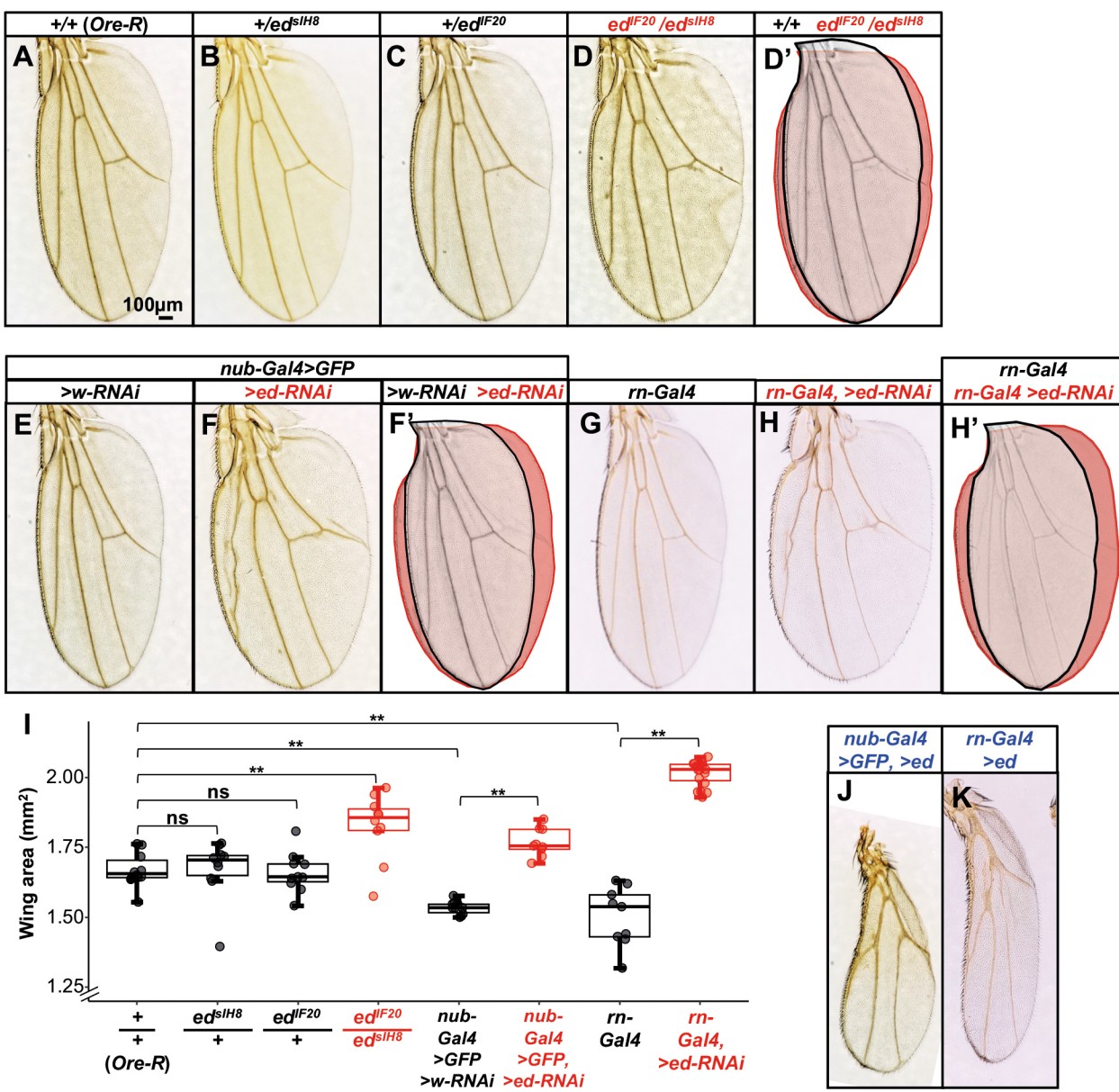

**Fig. 7. Reduced *echinoid* function increases adult wing size.** (A-I) Effect of reducing *ed* function on adult wing size. (A-H′) Adult wings of the indicated genotypes. (D′) Overlay of A and D. (F′) Overlay of E and F. (H′) Overlay of G and H. Quantification of wing areas is shown in I. *n*=10 wings [+/+ (*Oregon-R*); *ed*$^{slH8}$/+; *ed*$^{IF20}$/+; *ed*$^{slH8}$/*ed*$^{IF20}$; *nub-Gal4, >GFP, >w-RNAi*], 9 wings (*nub-Gal4, >GFP, >ed-RNAi; rn-Gal4*) and 19 wings (*rn-Gal4, >ed-RNAi*). ns indicates *P*>0.05, **P*<0.01 (one-way ANOVA with post-hoc Tukey's HSD test). For box and whisker plots, the horizontal line is the median, the box is the interquartile range and the whiskers extend to the largest (upper whisker) or smallest (lower whisker) value that is no further from the hinge than 1.5× inter-quartile range. (J,K) Effect of overexpressing *ed* on adult wing size. Overexpression using either *nub-Gal* (J) or *rn-Gal4* (K) dramatically reduced wing size. Scale bar in A applies to A-H′,J,K.

*rn-Gal4* to express *UAS-ed-RNAi* and found that this increased wing area by 15% (*nub-Gal4*) and 34% (*rn-Gal4*) compared to controls (Fig. 7E-H,H′,I). Conversely, overexpression of *ed* using *nub-Gal4* or *rn-Gal4* resulted in smaller wings (Fig. 7J,K). This may indicate that wing size is directly sensitive to *ed* dosage, but it is also possible that different mechanisms mediate *ed* overexpression and depletion phenotypes (e.g. *ed* overexpression could reduce wing size through non-specific toxicity). Because the *nub-Gal4* and *rn-Gal4* lines have reduced wing size compared to *Oregon-R* (Fig. 7I), all wing-area experiments using *Gal4* drivers were compared to controls that also contained the same *Gal4* driver, rather than to *Oregon-R*, to minimize confounding effects due to genetic background. Thus, as described previously (Bai et al., 2001; Yue et al., 2012), reducing

functional *ed* results in tissue overgrowth whereas overexpressing *ed* leads to reduced tissue size.

Ed could function in cell-cell adhesion, as a signaling molecule, or have both functions. To determine which of these properties of Ed are necessary for growth arrest, we examined how wing size is affected by expression of transgenes encoding Ed proteins that contain the extracellular and transmembrane domains but not the intracellular domain [*UAS-ed*$^{ΔC-GFP}$ (Laplante and Nilson, 2011) or the less well characterized *UAS-ed*$^{ΔIntra-GFP}$ (a gift from J.-C. Hsu, National Tsing Hua University, Hsinchu, Taiwan)]. These truncated proteins would presumably retain the capacity to form extracellular adhesions, but not function as signaling molecules. Overexpression of *ed*$^{ΔC-GFP}$ reduced adult wing size compared to controls (Fig. S6A-B,H), but *UAS-*

$ed^{\Delta Intra\text{-}GFP}$ expression did not have a significant effect on wing size (Fig. S6C-D,H). The effect of overexpressing the full-length protein ($UAS\text{-}ed^{Full}$) was much stronger (Fig. 7J,K) but wing areas could not be quantified because most wings were crumpled. Next, we examined whether either $ed^{Full}$ or $ed^{\Delta C\text{-}GFP}$ can rescue the overgrowth phenotype of $ed^{sIH8}/ed^{IF20}$ wings (Fig. S6E-H). Inclusion of $nub\text{-}Gal4$ in the $ed^{sIH8}/ed^{IF20}$ background reduced wing sizes to a size similar to that of the wild-type and $ed/+$ flies (Fig. S6E,H). Including either a $UAS\text{-}ed^{Full}$ or $UAS\text{-}ed^{\Delta C\text{-}GFP}$ transgene further reduced wing size (Fig. S6F-H).

Ed-depleted wings have a rounded, stout shape due to a hexagonal packing defect during pupal wing elongation (Chan et al., 2021). We assessed the roundness of wings in our $ed$ depletion, overexpression and rescue experiments based on the ratio of proximodistal length to anteroposterior width (PD/AP ratio) (Fig. S4I). Consistent with previous reports, we found that $ed^{sIH8}/ed^{IF20}$ and $nub$- or $rn\text{-}Gal4$, $UAS\text{-}ed\text{-}RNAi$ wings were significantly rounder than controls, as were $+/ed^{sIH8}$ and $+/ed^{IF20}$ heterozygotes. In the $ed^{sIH8}/ed^{IF20}$ background, expression of $UAS\text{-}ed^{Full}$ using $nub\text{-}Gal4$ rescued the defect in the PD/AP ratio. Roundness of $nub$- or $rn\text{-}Gal4$, $UAS\text{-}ed^{Full}$ wings in a wild-type background could not be quantified since most wings were crumpled. $nub\text{-}Gal4$-driven overexpression of $ed^{\Delta C\text{-}GFP}$, but not $rn\text{-}Gal4$-driven overexpression of $ed^{\Delta Intra\text{-}GFP}$, reduced wing roundness. $nub\text{-}Gal4$-driven overexpression of $ed^{\Delta C\text{-}GFP}$ also partially rescued the roundness of $ed^{sIH8}/ed^{IF20}$ wings.

In summary, expression of either $ed^{Full}$ or $ed^{\Delta C\text{-}GFP}$ produces opposing effects on wing size and shape to $ed$ depletion, and these transgenes can rescue the overgrowth and roundness of $ed^{sIH8}/ed^{IF20}$ wings. Since at least one transgene lacking the intracellular domain functions in this way, this suggests that the putative signaling domain is not essential for the effects of Ed on wing size or shape; at least some functions of Ed in wing morphogenesis are likely mediated by cell-cell adhesion. While the growth reduction caused by the overexpression transgenes could result from non-specific effects on cell growth, this is less likely to explain the effect on the wing PD/AP ratio. Additionally, since the hypomorphic missense allele $ed^{sIH8}$ present in our rescue experiments presumably retains a functional C-terminal domain, we cannot formally exclude the possibility that this protein might form a heterodimer with the $Ed^{\Delta C\text{-}GFP}$ protein that functions in an unusual way as a signaling molecule (the heterodimer would contain one wild-type extracellular domain and a wild-type cytoplasmic domain, albeit on different molecules).

## DISCUSSION

To systematically assess the requirement of individual cell-surface proteins in regulating cell survival or proliferation in clones, we examined the effect of reducing the expression of 74 such genes. Although we did not verify the efficacy of the RNAi knockdown in each case, we found that for 66 genes, there was no obvious decrease in clone size or alteration of clone shape. Thus, most cell-surface proteins, at least individually, do not function to sustain cell survival or proliferation.

Of the genes identified in the screen, we chose to focus on $ed$ for two reasons. First, reducing $ed$ function has very different effects in clones and in entire tissues. Clones with reduced $ed$ function are eliminated, while imaginal discs with reduced $ed$ function in large regions overgrow. In some ways, this phenotype is reminiscent of cells with disruptions in apicobasal polarity, such as $scrib$ (Bilder et al., 2000; Brumby and Richardson, 2003; Hariharan and Bilder, 2006), with the important difference being that cells lacking $ed$ seem to preserve most aspects of their apicobasal polarity and their tissue architecture appears relatively normal.

Second, previous studies have reported that $ed$ mutant tissue has increased EGFR signaling (Bai et al., 2001; Fetting et al., 2009; Ho et al., 2010; Islam et al., 2003; Rawlins et al., 2003a; Spencer and Cagan, 2003) and reduced Hippo pathway activity, resulting in increased Yki-target gene expression (Yue et al., 2012). Both changes would be expected to promote cell survival and proliferation, thus the underrepresentation and elimination of $ed$ mutant clones is unexpected. Previous work (Chang et al., 2011; Laplante and Nilson, 2006, 2011; Lin et al., 2007; Wei et al., 2005) and our observations (Fig. S2) have shown that $ed$ mutant cells sort away from wild-type cells and an actomyosin cable forms in wild-type cells at the clone interface that could potentially promote clone extrusion. We also confirmed that EGFR signaling is likely elevated, as assessed by reduced levels of nuclear Capicua (Fig. S7A-C). However, our observations are not consistent with the previously proposed effect on the Hippo pathway (Fig. 5A-E, Fig. S4A-D).

A study that investigated neuroblast quiescence (Ding et al., 2016) seems, at first glance, to support the previously proposed view of Ed as a positive regulator of Hippo signaling. In that study, reducing Hippo signaling in the neuroblast could promote an emergence from quiescence, as assessed by an increase in neuroblast diameter and proliferation. Autonomous expression of $ed\text{-}RNAi$ also increased neuroblast size. However, neuroblast size increase was also observed when $ed\text{-}RNAi$ was expressed in adjacent glial cells. This effect could therefore also be explained by a reduction in adhesion of the neuroblast to its glial niche, which then allows for increased growth and proliferation. This result also highlights how more complex and non-cell-autonomous mechanisms could link Ed to the Hippo pathway.

## Why are *ed* clones eliminated?

Cells lacking $ed$ function have increased apoptosis. We observed this in clones, especially at clone boundaries, and when $ed$ was depleted in broader domains. This elevated baseline propensity for apoptosis in mutant cells, enhanced at interfaces with wild-type cells, resembles the phenotype resulting from $Minute$ mutations (Akai et al., 2021; Baumgartner et al., 2021; Coelho et al., 2005; Recasens-Alvarez et al., 2021). We found that $ed$ mutant tissue has reduced levels of the anti-apoptotic protein Diap1 and that $ed$ clone elimination could be rescued by co-expressing $diap1$ with $ed\text{-}RNAi$. These observations suggest the reduction in Diap1 levels underlies, in significant part, $ed$ clone elimination.

In $ed$ clones, we observed decreased expression of a $diap1$ transcriptional reporter that was constructed using multiple copies of a Hippo-responsive element from the $diap1$ locus (Wu et al., 2008). These data indicate that the change in Diap1 protein levels is due, at least in part, to decreased Yki-dependent $diap1$ transcription. We also found decreased expression of $ban\text{-}lacZ$ and $fj\text{-}lacZ$ in $ed$ mutant clones but no change in nuclear Yki localization. These findings differ from a previous study reporting that nuclear Yki in $ed$ clones increases Diap1 levels and expression of other Yki-target genes (Yue et al., 2012). Thus, the regulation of Yki-target genes by Ed is not easily explained by a simple effect on alterations in Hippo pathway activity.

Often, wild-type cells directly adjacent to $ed$ clones have elevated expression of Yki-target genes and elevated Diap1 protein. In this way, these cells resemble cells with Hippo mutations pathway that behave as supercompetitors (Tyler et al., 2007). Thus, Diap1 reduction in $ed$ mutant clones, coupled with the generation of boundary cells that have some properties of supercompetitors, could together contribute to the elimination of $ed$ cells from the disc by apoptosis (Fig. 8A,A').

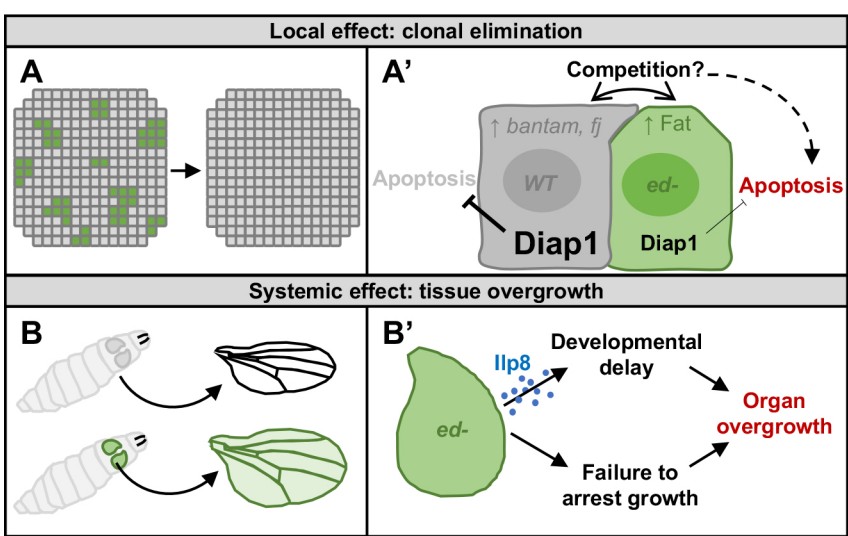

**Fig. 8. Model summary of phenotypes caused by Echinoid loss and proposed mechanisms.**
(A,A′) Clones of *echinoid* mutant cells are eliminated from mosaic tissues. Ed-depleted cells have decreased Diap1 expression, which predisposes them to death by apoptosis, and increased apical Ft. In the wild-type neighbors bordering the *ed* clones, levels of Yki targets, including Diap1, *bantam* and *four-jointed*, are elevated. This may confer a competitive advantage to the wild-type neighbors, which could facilitate the elimination of *ed* clones from mosaic tissue (A′). (B,B′) When *ed* mutant tissue is abundant (e.g. in an entire organ), mutant cells persist and the resulting organs overgrow. This overgrowth is facilitated by Ilp8 secretion, which delays pupariation. Slow-growing *ed* mutant tissue fails to arrest at the proper final size, leading to overgrown organs (B′).

## Why are *ed* mutant tissues overgrown?

Although *ed* tissue grows more slowly than wild-type tissue (likely due to elevated rates of apoptosis), it appears to grow well beyond an appropriate size. This cannot simply result from the delay in pupariation caused by Ilp8 production by *ed* tissue, since delaying pupariation by reducing ecdysone production still results in a growth arrest at close to the appropriate size (Parker and Struhl, 2020). Thus, Ed has a second function in arresting growth at the appropriate final size (Fig. 8B,B′).

Ed can function as an adhesion molecule. The levels of E-cadherin are known to be altered in *ed* mutant tissue and Ed and E-cadherin may function with some degree of redundancy to mediate cell-cell adhesion (Chang et al., 2011; Laplante and Nilson, 2006; Wei et al., 2005). The mechanisms by which tissues sense their final size and arrest their growth accordingly are still poorly understood. While growth arrest is clearly disrupted in mutants that completely disrupt apicobasal polarity (reviewed by Hariharan and Bilder, 2006), it is possible, even likely, that subtle changes in junctional components could disrupt growth arrest.

## Concluding remarks

While our studies have identified two clearly separable roles of Ed in regulating tissue growth, neither function can be easily attributed to the effect of Ed on a single signaling pathway. Ed participates in, and regulates, at least two key cellular processes: adhesion at the adherens junction (Chang et al., 2011; Laplante and Nilson, 2006, 2011; Wei et al., 2005); and endocytosis and endosomal trafficking (Chan et al., 2021; Fetting et al., 2009; Ho et al., 2010; Li et al., 2015; Rawlins et al., 2003b; Yang et al., 2018). Moreover, at least based on genetic interactions, *ed* influences EGFR and Notch signaling, and likely has a complex interaction with outputs of the Hippo pathway, such as regulating Diap1 levels. Ed may interact with more pathways or processes than we currently appreciate, either directly or indirectly via its effects on adhesion, endomembrane trafficking or through signaling crosstalk. The collection of phenotypic abnormalities and alterations in gene expression that we observe in *ed* mutants likely involve a summation of alterations in multiple pathways rather than a disruptive effect on any single pathway.

## MATERIALS AND METHODS
### Fly stocks and husbandry

Unless otherwise noted, all experimental crosses were raised at 25°C on food prepared according to the recipe from the Bloomington *Drosophila*

Stock Center. *Drosophila melanogaster* stocks used in this study include or were derived from the following: *Oregon-R* ('*Ore-R*', used as wild type); *y w hs-FLP; act<[y+]<Gal4 UAS-GFP/SM5-TM6B*, *hs-FLP;; act<stop<Gal4 UAS-RFP/SM5-TM6B*, *TIE-DYE* (Worley et al., 2013); *FRT40A* and *FRT40A, white+ ubi-GFP* (Xu and Rubin, 1993); *FRT40A MARCM* (Lee and Luo, 1999); *eyFLP; FRT40A CL white+/CyO* (BL5622); *UAS-ed* (Bai et al., 2001); *ed$^{IF20}$ FRT40A*, *ed$^{1x5}$ FRT40A* and *ed$^{sIH8}$ FRT40A* (Bai et al., 2001); *UAS-ed$^{Full}$* and *UAS-ed$^{\Delta C}$-GFP* (Laplante and Nilson, 2011); *UAS-ed$^{\Delta Intra-GFP}$* (a gift from J.-C. Hsu); *Yki:GFP* (Fletcher et al., 2018); *nub-Gal4* (*AC-62*, BL25754); *rn-Gal4* (BL7405); *hh-Gal4* (BL45169); *en-Gal4* (BL25752); *ban-lacZ* (BL10154) (Herranz et al., 2012); *fj-LacZ* (*ff$^{P1}$*, BL44253) (Brodsky and Steller, 1996); *ex-lacZ* (Boedigheimer and Laughon, 1993); *diap1-GFP.HREx8* (Wu et al., 2008); *dIlp8$^{MI00727}$*(BL33079); *UAS-w-RNAi* (BL33644); *UAS-ama-RNAi* (BL33416); *UAS-beat-Vc-RNAi* (BL60067); *UAS-Cont-RNAi* (BL34867); *UAS-shg-RNAi* (BL32904); *UAS-side-VII-RNAi* (V10011); *UAS-ft-RNAi* (BL34970); *UAS-ds-RNAi* (BL32964); *UAS-ed-RNAi* [V104279, V3087, V938 and BL38423 ('*ed-RNAi*' refers to V104279 unless otherwise indicated)]; *UAS-otk2-RNAi* (BL55892); and *UAS-p35* (BL5073). 'BL' and 'V' indicates stocks obtained from the Bloomington *Drosophila* Stock Center (BDSC; Bloomington, IN, USA) and Vienna *Drosophila* Resource Center (VDRC; Vienna, Austria), respectively. Additional stocks which were included in the genetic screen but are not mentioned in the main text are listed in Table S3 with BDSC or VDRC numbers indicated.

### Mosaic tissue generation

Clones induced by heat shock were generated in a 37°C water bath 48 h before dissection, unless otherwise noted. FLP-out Gal4 clones were made using heat shocks of 6 min (to generate clones at low density), 12 min (for moderately low density), 15 min (for medium density) or 30 min (for high density). MARCM clones were generated 72 h AEL using a 1 h heat shock. Mitotic recombination clones were generated at indicated time points using a 1 h heat shock. Mitotic recombination clones made in the eye were induced by expression of the *eyFLP* driver.

### Screen

Approximately 10 *UAS-RNAi* males were crossed to ~20 *y, w, hs-FLP; act<[y+]<Gal4, UAS-GFP/SM5-TM6B* or *TIE-DYE (Act<stop<lacZ.nls, Ubi <stop<GFP.nls; Act<stop<GAL4, UAS-his2A::RFP/SM5-TM6B)* virgin females. Crosses were kept on Bloomington food supplemented with yeast and flipped once daily. Clones were induced by heat shock on day 3. Early rounds of screening used a 15 min heat shock, although we later switched to a 12 min heat shock for the majority of the screen since the low clone density made identifying deviations from the control in either direction easier. Wing imaginal discs from ~6 wandering L3 larvae per line were dissected ~48 h after heat shock, stained with DAPI and imaged.

## Immunohistochemistry and fluorescence microscopy

Imaginal discs were dissected in PBS, fixed in 4% paraformaldehyde in PBS and permeabilized in 0.1% Triton in PBS. When staining discs with anti-Yorkie, the discs were instead dissected in 0.1 M NaPO$_4$, fixed in PLP fixative (2% PFA, 0.01 M NaIO$_4$, 0.075 M Lysine and 0.037 M NaPO$_4$) and permeabilized in 10% NGS in Saponin+NaPO$_4$. Primary antibody incubations were carried out overnight at 4°C. Secondary antibody incubations were carried out for 2-3 h at room temperature or overnight at 4°C. Discs were mounted in SlowFade Diamond Antifade Mountant (S36963, Invitrogen).

Primary antibodies used were: rabbit anti-Ed (1:500, a gift from J.-C. Hsu) (Wei et al., 2005), rabbit anti-cleaved Dcp-1 (1:250; Cell Signaling Technology, 9578, RRID:AB_2721060), mouse anti-Diap1 (1:200, a gift from B. Hay, California Institute of Technology, Pasadena, CA, USA), mouse anti-β-Galactosidase (1:500, Sigma-Aldrich, WH0051083M1, RRID: AB_1841716), mouse anti-β-Galactosidase (1:500, SAB4200805, Sigma-Aldrich), chicken anti-GFP (1:500, Abcam, ab13970, RRID:AB_300798), guinea pig anti-Yorkie (1:500, I.K.H. - UC Berkeley 1798, RRID: AB_3711287), rat anti-Fat (1:400, a gift from K. Irvine, Rutgers University, NJ, USA) (Feng and Irvine, 2009), rabbit anti-pMAD (1:400, Abcam, ab52903, RRID:AB_882596) and guinea pig anti-Cic (1:300, I.K.H. - UC Berkeley 1503, RRID: AB_3711286) (Tseng et al., 2007). Secondary Alexa Fluor antibodies used were: goat anti-chicken IgG Alexa Fluor 488 (1:500; Abcam, ab150169, RRID:AB_2636803), goat anti-rat IgG Alexa Fluor 555 (1:500; Thermo Fisher Scientific, A-21434, RRID:AB_2535855), goat anti-mouse IgG Alexa Fluor 555 (1:500; Thermo Fisher Scientific, A-21422, RRID:AB_2535844), goat anti-guinea pig IgG Alexa Fluor 647 (1:500; Thermo Fisher Scientific, A-21450, RRID:AB_2535867) and goat anti-rabbit IgG Alexa Fluor 647 (1:500; Thermo Fisher Scientific, A32733, RRID: AB_2633282). Nuclei were stained with DAPI (1:1000, Cell Signaling).

Fluorescence images were taken on a Zeiss Axio Imager M2 equipped with a 20× objective (Plan-Apochromat, 20×/0.8), LED light source (Excelitas Technologies), AxioCam 506 mono camera (Zeiss) and ApoTome.2 slider for optical sectioning. Images and image stacks were acquired and optically sectioned in ZEN 2.3 software (Zeiss). Images were processed using FIJI software (Schindelin et al., 2012). Unless otherwise noted, images show a single *z*-plane.

## Adult wing imaging and quantification

Adult wings were dissected from female flies. One wing per fly was mounted in Gary's Magic Mountant (Lawrence et al., 1986). Wings were imaged using a Keyence VHX-5000 digital microscope, using the 20-200× lens at 150×. Brightness, contrast and color tone of wing images have been adjusted on some images for improved visibility of features relevant to this study (wing shape and size).

For qualitative comparisons of wing sizes, wing images or traced silhouettes were overlaid in Microsoft PowerPoint. For quantitative comparisons of wing sizes, wings were traced and area was quantified in FIJI (Schindelin et al., 2012). Charts were generated using the ggplot2 package in Rstudio (Wickham, 2016).

Wing aspect ratios were calculated by dividing the length of proximodistal (PD) axis (measured from the posterior junction of the wing and hinge to the tip of the L3 vein) by the length of the anteroposterior (AP) axis (measured as the shortest distance from the tip of the L5 vein to the L1 margin).

## Statistical analysis

$P$ values were obtained by one-way ANOVA with Tukey's HSD test using Astatsa freeware (https://astatsa.com). $P<0.05$ was considered significant. For box and whisker plots, the horizontal line is the median, the box is the interquartile range and the whiskers extend to the largest (upper whisker) or smallest (lower whisker) value that is no further from the hinge than 1.5× inter-quartile range.

## Acknowledgements

We thank current and former members of the Hariharan laboratory for helpful feedback and discussions, especially Luigi Viggiano, who conducted related experiments that influenced our thinking on this study, and Melanie Worley. We thank Zohra Allata for help with dissections and microscopy. We thank FlyBase (https://flybase.org) for invaluable curation of information. We also thank Bruce Hay, Jui-Chou Hsu, Ken Irvine, Laura Nilson, Duojia Pan, Nic Tapon, the Bloomington *Drosophila* Stock Center, the Vienna *Drosophila* Resource Center and the Developmental Studies Hybridoma Bank for reagents and fly stocks. Some data presented in this work have been previously presented in a publicly available PhD thesis (Spitzer, 2023).

## Competing interests

The authors declare no competing or financial interests.

## Author contributions

Conceptualization: D.C.S., I.K.H; Data curation: D.C.S.; Formal analysis: D.C.S.; Funding acquisition: I.K.H.; Investigation: D.C.S., W.Y.S., A.R.-V.; Supervision: I.K.H.; Visualization: D.C.S.; Writing – original draft: D.C.S., I.K.H.; Writing – review & editing: D.C.S., I.K.H.

## Funding

This work was funded by the National Institutes of Health (R35 GM122490 to I.K.H.) and by a pre-doctoral fellowship from the National Science Foundation (DGE 1752814 to D.C.S.). Open Access funding provided by the University of California. Deposited in PMC for immediate release.

## Data and resource availability

All relevant data can be found within the article and its supplementary information.

## The people behind the papers

This article has an associated 'The people behind the papers' interview with some of the authors.

## Peer review history

The peer review history is available online at https://journals.biologists.com/dev/lookup/doi/10.1242/dev.204572.reviewer-comments.pdf

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
