## [Peer Review File · Development (Cambridge, England)]

The cell adhesion molecule Echinoid promotes tissue survival and separately restricts tissue overgrowth

Danielle C. Spitzer, William Y. Sun, Anthony Rodríguez-Vargas and Iswar K. Hariharan
DOI: 10.1242/dev.204572

Editor: Swathi Arur

Review timeline

Original submission:	2 December 2024
Editorial decision:	23 December 2024
First revision received:	23 June 2025
Accepted:	7 July 2025

Original submission

First decision letter

MS ID#: dev.204572

MS Title: The cell adhesion molecule Echinoid promotes tissue survival and separately restricts tissue overgrowth

Authors: Danielle Spitzer; William Sun; Anthony Rodríguez-Vargas; Iswar Hariharan
Article Type: Research Article

Dear Dr Hariharan,

I have now received all the referees' reports on the above manuscript, and have reached a decision. The referees' comments are appended below, or you can access them online: please go to:

As you will see, the referees are in agreement with the significance of the work and provide recommendations to improve the clarity and rigor of the analysis. If you are able to revise the manuscript along the lines suggested, which may involve further experiments, I will be happy receive a revised version of the manuscript. Your revised paper will be re-reviewed by one or more of the original referees.

Please attend to all of the reviewers' comments and ensure that you clearly highlight all changes made in the revised manuscript. Please avoid using 'Tracked changes' in Word files as these are lost in PDF conversion. I should be grateful if you would also provide a point-by-point response detailing how you have dealt with the points raised by the reviewers in the 'Response to Reviewers' box. If you do not agree with any of their criticisms or suggestions please explain clearly why this is so.

Reviewer 1

SUMMARY OF THE ADVANCE MADE IN THIS PAPER AND ITS POTENTIAL SIGNIFICANCE TO THE FIELD

The present manuscript by Spitzer et al examines the role of the Ig superfamily protein Echinoid (Ed) in tissue growth in *Drosophila*. In a screen for cell adhesion molecules that affect clone shape when knocked down in genetic mosaics, the authors show that ed depletion leads to cell elimination at cell boundaries in a process akin to cell competition. In contrast to previous

assertions that Ed acts as an upstream activator of the Hippo pathway, the authors show that Yorkie target genes are generally repressed in ed mutant clones, while they are upregulated at the clone boundary, possibly explaining how ed-depleted cells are eliminated. Compartment-wide knockdown of Ed leads to upregulation of dilp8 and consequent delay in the larval/pupal transition, as well as increased wing disc size. This study adds to our understanding of tissue size control both by clarifying the relationship between Ed and Hippo signalling, and by providing new mechanistic insight into the overgrowth induced by tissue-wide ed depletion. I would recommend publication in Development provided the points below are addressed.

SUGGESTIONS TO AUTHORS

Major points

1. The change in Fat levels in and around ed clones in Figure 5F-F' is an interesting finding as it suggests a potential mechanism for Yki target upregulation around the clone boundaries. To properly document this result, the authors should quantify Fat localisation using another membrane protein as a control in and around ed clones and control clones.
2. In Figure 6E-K, the authors show that disc size increases in hh>ed-RNAi animal and suggest that this indicates a role for Ed in wing size control that cannot be explained by Dilp8 expression alone as Parker and Struhl have shown that delaying the larval/pupal transition alone does not increase wing disc size. In Figure 6-G-K, it looks like both the anterior compartment (where ed is not knocked down) and the posterior compartment (where it is) are overgrowing. Could the authors comment on why this may be the case? One possibility is that the cell death observed in the ed-depleted compartment contributes to this overgrowth by promoting a regenerative programme. Perhaps the authors could block this using Dronc-DN or RHG depletion (to avoid the pitfalls of p35 overexpression driving compensatory proliferation) to test if this is the case? Are other markers of regeneration such as JNK or Upd3 also elevated?

Minor points

1. The first paragraph of the introduction should be referenced.
2. MARCM clones in Figure 2: please indicate when the heat shocks were carried out.
3. Figure S3: what were the conditions used for this experiment and why are the clones a lot larger than in Figure 2?
4. Line 487-8 "...although one target, ex-lacZ, was indeed increased..." I could not see this experiment in the manuscript. There is ex-LacZ upon ed overexpression (Fig. 5F) but not loss of function as far as I could see.

Reviewer 2

SUMMARY OF THE ADVANCE MADE IN THIS PAPER AND ITS POTENTIAL SIGNIFICANCE TO THE FIELD

The manuscript by Spitzer et al examines cell competition in Drosophila epithelia tissues, which is an important biological process that ensures proper organ growth during development and organism fitness. Given that cell competition involves signalling between neighbouring cells, the authors theorise that additional cellular adhesion molecules exist that mediate this. They investigate this by performing an in vivo RNAi screen for most cell adhesion proteins in the Drosophila genome and identify a handful that affect clone growth and shape. Subsequently, they focus on the immunoglobulin domain containing protein Echinoid and report three main findings, which are all important and novel:

- Echinoid mutant clones generated in mosaic epithelial tissues (wing imaginal discs) undergrow and appear to be subject to cell competition.
- Epithelial tissues that are comprised of only Echinoid mutant tissue (wing or eye imaginal discs) overgrow, probably because they induce expression of the Dilp8 hormone, which subsequently induces an extended growth period for the organism.
- Echinoid mutant cells do not possess modulated Hippo/Yorkie activity. This latter finding questions the validity of previous studies that reported that Echinoid was an upstream member of the Hippo pathway.

As such, the manuscript reports several pieces of data that will be of value to scientists that study cell competition, epithelial tissue growth and cell signalling (especially Hippo signalling). The manuscript is somewhat different to most papers that report an in-depth characterisation of one

phenotype. Instead, it presents three different but related messages, that will set the scene for further studies that characterise exactly how Echinoid impacts clone growth in a mosaic setting and organ growth in a non-mosaic setting. Importantly, it also revises our understanding of Hippo signalling.

The manuscript represents a large body of work, especially the genetic screen, which would have been laborious to execute. The data are of a high quality and include appropriate controls and statistical analyses and therefore are very convincing. Importantly, this will mean that other groups can readily build on the manuscript as a reliable and valuable source of information.

SUGGESTIONS TO AUTHORS

The authors attempted to determine whether Echinoid influences clone growth via a cell adhesion function or a signalling function, using a transgenic rescue approach, although the results were not completely conclusive. Tailored Echinoid CRISPR alleles are probably the best way to approach this, but this is not trivial. Instead, perhaps the authors could use an alternate wing disc Gal4 driver that does not have a phenotype on its own (like nubbin-Gal4 did). I agree with the authors that Echinoid probably influences clone growth via a cell adhesion function, especially given that Echinoid and E-cad LOF clones phenotypically resemble each other, but it could be tested more rigorously.

To make the manuscript easier to read, the authors could streamline and shorten the Results and Discussion sections.

Reviewer 3

This manuscript explores the functions of Echinoid (Ed) and other cell adhesion molecules in organ growth. Ed has been implicated in the regulation of several signaling pathways, including EGFR, Notch, and Hippo, and appears to be involved in complex molecular interactions that may relate to endomembrane trafficking, cell adhesion, and possibly signaling. Careful genetic experiments are presented that strive to explain paradoxical outcomes of ed loss: small clones of cells that are mutant for ed are eliminated via apoptosis, whereas large areas of ed mutant tissue exhibit overgrowth. The first outcome is attributed to a reduction in Diap1 and a concomitant increase in apoptosis in small ed clones. Surprisingly, surrounding areas show increased Diap1, creating a possibility for cell competition. The second outcome may stem from an extension of the larval stage due to hyperactivation of Iip8 and possibly other unknown mechanisms that contribute to a failure to arrest growth.

Some of the reported results are in direct contradiction with a previously published study (Yue et al., 2012). While some of these contradictions can be simply explained by the other study using a cell lethal mutation that favored the overgrowth of ed mutant tissue, others cannot be explained in a simple way, such as the discrepancy in Diap1 expression and Yki nuclear localization in ed mutant cells. Overall, this manuscript proposes interesting, if not provocative, new models for explaining the complex functions of Ed in development and is written very well. With rigorous data and thoughtful analysis, it will contribute to a continued discourse in the field by highlighting the complexities surrounding Ed functions.

Major suggestions for improvement:

At present, it is difficult to evaluate the significance of the involvement of Diap1 in the survival of ed mutant cells shown in Fig. 4G-H', for two reasons. First, the difference between H and G is not as striking as some of the other results; second, since ed clone survival appears to be rather sensitive to the duration and timing of heat shock (Fig. 3), the result shown in H might be attributed to a fluctuation in experimental conditions. A more solid demonstration of Diap1 involvement in ed-mediated apoptosis may be obtained e.g. via quantification of the differences in clone number or size. Another possibility is to repeat this experiment using low clone density, as the differences may become more apparent (like the comparison in Fig. 4D vs. 4C).

There is a discrepancy between nuclear Yki localization reported in ed mutant cells in Yue et al. vs. no changes observed in this manuscript. Yue et al. used antibody staining for Yki, whereas this

manuscript used a well-validated CRISPR-generated line where Yki is tagged with GFP. I suggest checking Yki localization using antibody staining in addition to using this line, to confirm this result. Also, since Yki target genes show higher expression in cells surrounding the clones, it is puzzling why Yki levels would not show an increase in those cells. The Fat explanation would still require increased Yki nuclear localization as a downstream outcome, so higher nuclear Yki would be expected in surrounding cells. It may also be possible to explain the induction of Yki target genes in surrounding cells through decreased Hpo signaling due to mechanical effects stemming from actomyosin cable formation in those cells. But this latter explanation would still expect an increase in nuclear Yki.

Minor points:

Fig. 2J,K should indicate in the figure itself and/or in the legend which cell lethal mutation was used (gene and allele).

Check sentence starting at line 381 ("We also reduced..."), looks like "to" is missing before "drive".

In Fig. 6B', apoptosis is observed throughout the area where *ed* is knocked down. How can this result be reconciled with Fig. 4F', where apoptosis was mostly observed at the edges of the clones, and with a model where apoptosis is driven by the difference in Diap1 levels in adjacent cells?

First revision

Author response to reviewers' comments

We thank the Reviewers for their time and helpful feedback. We have added new data, improved figure panels and layouts, and made textual changes. Our most significant changes include: a more rigorous examination of Yki localization in and around *ed* clones, and a more nuanced discussion about the mechanism that may mediate the "border effect," addition of a new supplemental figure showing pMad expression in overgrown discs with *ed*-depletion in one compartment, and a deeper, more extensive analysis of the phenotypes associated with expression of full-length or intracellularly-truncated *ed* transgenes using multiple drivers and alleles. A point-by-point description of our responses to Reviewers' comments is below.

Reviewer 1

SUMMARY OF THE ADVANCE MADE IN THIS PAPER AND ITS POTENTIAL SIGNIFICANCE TO THE FIELD

The present manuscript by Spitzer et al examines the role of the Ig superfamily protein Echinoid (Ed) in tissue growth in Drosophila. In a screen for cell adhesion molecules that affect clone shape when knocked down in genetic mosaics, the authors show that ed depletion leads to cell elimination at cell boundaries in a process akin to cell competition. In contrast to previous assertions that Ed acts as an upstream activator of the Hippo pathway, the authors show that Yorkie target genes are generally repressed in ed mutant clones, while they are upregulated at the clone boundary, possibly explaining how ed-depleted cells are eliminated. Compartment-wide knockdown of Ed leads to upregulation of dilp8 and consequent delay in the larval/pupal transition, as well as increased wing disc size. This study adds to our understanding of tissue size control both by clarifying the relationship between Ed and Hippo signalling, and by providing new mechanistic insight into the overgrowth induced by tissue-wide ed depletion. I would recommend publication in Development provided the points below are addressed.

Response: We thank Reviewer 1 for their feedback and recommendation for publication of our work, provided that we satisfactorily address their comments. We have taken every effort to do so.

SUGGESTIONS TO AUTHORS

The change in Fat levels in and around ed clones in Figure 5F-F' is an interesting finding as it suggests a potential mechanism for Yki target upregulation around the clone boundaries. To properly document this result, the authors should quantify Fat localisation using another membrane protein as a control in and around ed clones and control clones.

Response: We thank the Reviewer for this suggestion. We repeated the experiment and included an antibody stain against Dlg, which localizes to the lateral membrane abutting the adherens junctions (we chose not to use a marker specific to the adherens junctions or apical complex since loss of Ed has been previously shown to alter the localization and/or levels of AJ/apical proteins such as e-Cadherin, Armadillo and Bazooka [Wei et al., 2005]). Consistent with previous observations, we found that the apical profiles of cells expressing *ed-RNAi* are smaller and that the wild-type cells immediately adjacent to them have larger apical profiles than other wild-type cells - this has been previously noted by others. The apparent reduction in Fat levels in neighbouring cells could simply be because of the dilution of the same amount of Fat being spread over a larger apical domain. This, together with the difficulty of finding a “control” protein that co-localizes with Ed and yet is unaffected by a loss of Ed makes these differences difficult to interpret even if quantified carefully. We have therefore re-written this portion of the text with a more conservative interpretation:

“Border effects such as that observed in *ed* clones have been described previously with manipulations of the Fat/Dachsous (Ft/DS) pathway (Matakatsu and Blair, 2012; Willecke et al., 2008). To investigate a possible role for this pathway, we stained discs containing *ed-RNAi* clones with anti-Ft. We observed increased staining in the clones (Fig. 5F, F') which might be due to increased apical Ft levels, concentration of Ft protein into a smaller area of apically-constricted membrane, or a combination of both. Apical constriction within *ed* clones, as well as apical expansion of their immediate neighbors, has been noticed previously (Chang et al., 2011; Laplante and Nilson, 2006; Wei et al., 2005). Apical expansion could dilute the Ft concentration in the cells immediately adjacent to the clone. A relative reduction in Ft levels in the border cells would be expected to result in reduced Hippo pathway activity (Bennett and Harvey, 2006; Cho et al., 2006; Silva et al., 2006; Willecke et al., 2006) and increased expression of Yki-target genes in those border cells as we have observed, thus providing a potential mechanistic explanation for the ways in which cells with reduced *ed* function interact with their wild-type neighbors. In the absence of any direct evidence that the increased expression of Yki target genes in wild-type cells bordering the clone is caused by alterations in Ft distribution, a variety of other mechanisms ranging from changes in various signaling pathways to alterations in cell adhesion could apply.”

In Figure 6E-K, the authors show that disc size increases in hh>ed-RNAi animal and suggest that this indicates a role for Ed in wing size control that cannot be explained by Dilp8 expression alone as Parker and Struhl have shown that delaying the larval/pupal transition alone does not increase wing disc size. In Figure 6-G-K, it looks like both the anterior compartment (where ed is not knocked down) and the posterior compartment (where it is) are overgrowing. Could the authors comment on why this may be the case? One possibility is that the cell death observed in the ed-depleted compartment contributes to this overgrowth by promoting a regenerative programme. Perhaps the authors could block this using Dronc- DN or RHG depletion (to avoid the pitfalls of p35 overexpression driving compensatory proliferation) to test if this is the case? Are other markers of regeneration such as JNK or Upd3 also elevated?

Response: Thank you for this comment. We have examined a number of discs where *ed-RNAi* is expressed in the posterior compartment. We observe enlarged imaginal discs where, in each case, the anterior compartment is also enlarged. We think that this is unlikely to be due to regenerative growth because we do not observe excessive growth only near the compartment boundary but over the entire disc. When we examine the pattern of phospho-Mad staining, it looks relatively physiological with the phospho-Mad stripe being proportionally larger in the overgrown discs. A likely explanation is that an enlarged posterior compartment which has not arrested its growth makes more Hedgehog and that this results in increased Dpp production and increased growth of the entire disc. This is reminiscent of the phenomenon of “accommodation” described by Garcia-Bellido and colleagues where some kinds of growth alterations in one compartment appear to induce corresponding changes in the other (which Garcia-Bellido used to argue against the

notion of strict growth autonomy of compartments). We have added in the following text, as well as a new Supplemental Figure showing examples of phospho-Mad staining of wild-type and *en>ed-RNAi* discs:

“Interestingly, although the *ed-RNAi* is only expressed in the posterior compartment, increased growth was also observed in the anterior compartment. This observation is consistent with the phenomenon of positional accommodation that can occur in imaginal discs, where changes to growth in mosaic tissues exerts a non-autonomous effect on the growth of wild-type tissues (Díaz-Benjumea et al., 1989; García-Bellido, 2009; García-Bellido et al., 1994). This may be mediated by increased morphogen production from the enlarged posterior. We observed a relatively normal pattern of phospho-Mad staining in overgrown discs generated using *en-Gal4* (which has a similar expression pattern to *hh-Gal4*), indicating that Dpp production scales with the larger discs (Fig. S5).”

The first paragraph of the introduction should be referenced.

Response: Thank you for this suggestion. We have added the following references to the first paragraph:

Loyd, 2013
 Conlon and Raff, 1999
 LeGoff and Lecuit, 2016
 Matamoro-Vidal and Levayer, 2019
 Bielmeier et al., 2016
 Grata and Levayer, 2025

MARCM clones in Figure 2: please indicate when the heat shocks were carried out.

Response: Thank you; we have added the following language to the Methods section: “MARCM clones were generated 72 h after AEL using a 1-hr heat shock.”

Figure S3: what were the conditions used for this experiment and why are the clones a lot larger than in Figure 2?

Response: We appreciate this comment and recognize that the discrepancy in clone size may raise questions from our readers. Because the goal of this experiment was simply to evaluate Xrp1 levels in *ed* clones v. surrounding wild type tissue—not to draw

conclusions about clone size, shape, or survival—we did not use as stringent timing controls as we used in other experiments. As expected, this resulted in heterogeneity in clone size and recovery in the discs we dissected; we selected an image that had relatively large and numerous clones since it was the single image that most clearly illustrated the result. The results were clear enough to inform our decision to not conduct additional experiments examining Xrp1. We felt that would be valuable to our readers to include as a supplemental figure.

To avoid confusion and to clarify to readers that they should not draw unfounded conclusions based on the size of the clones in this experiment, we added the following language to the legend:

“Timing of clone generation and larval dissection were not as tightly controlled as in experiments intended to assess clone survival or morphology.”

Line 487-8 "...although one target, ex-lacZ, was indeed increased..." I could not see this experiment in the manuscript. There is ex-LacZ upon ed overexpression (Fig. 5F) but not loss of function as far as I could see.

Response: We thank the Reviewer for catching this error. This refers to an experiment that we included in an earlier version of the manuscript (shared as a preprint on bioRxiv: *Spitzer et al., 2023*), where we examined *ex-lacZ* in flies where *ed* was knocked down in the posterior compartment. When revising and reorganizing our results to present a clearer story, we removed that experiment but failed to remove the reference to it in the main text. We apologize for this error and have removed the reference from our revised manuscript.

Reviewer 2

SUMMARY OF THE ADVANCE MADE IN THIS PAPER AND ITS POTENTIAL SIGNIFICANCE TO THE FIELD

The manuscript by Spitzer et al examines cell competition in Drosophila epithelia tissues, which is an important biological process that ensures proper organ growth during development and organism fitness. Given that cell competition involves signalling between neighbouring cells, the authors theorise that additional cellular adhesion molecules exist that mediate this. They investigate this by performing an in vivo RNAi screen for most cell adhesion proteins in the Drosophila genome and identify a handful that affect clone growth and shape. Subsequently, they focus on the immunoglobulin domain containing protein Echinoid and report three main findings, which are all important and novel:

- Echinoid mutant clones generated in mosaic epithelial tissues (wing imaginal discs) undergrow and appear to be subject to cell competition.*
- Epithelial tissues that are comprised of only Echinoid mutant tissue (wing or eye imaginal discs) overgrow, probably because they induce expression of the Dilp8 hormone, which subsequently induces an extended growth period for the organism.*
- Echinoid mutant cells do not possess modulated Hippo/Yorkie activity. This latter finding questions the validity of previous studies that reported that Echinoid was an upstream member of the Hippo pathway.*

As such, the manuscript reports several pieces of data that will be of value to scientists that study cell competition, epithelial tissue growth and cell signalling (especially Hippo signalling). The manuscript is somewhat different to most papers that report an in-depth characterisation of one phenotype. Instead, it presents three different but related messages, that will set the scene for further studies that characterise exactly how Echinoid impacts clone growth in a mosaic setting and organ growth in a non-mosaic setting. Importantly, it also revises our understanding of Hippo signalling.

The manuscript represents a large body of work, especially the genetic screen, which would have been laborious to execute. The data are of a high quality and include appropriate controls and statistical analyses and therefore are very convincing. Importantly, this will mean that other groups can readily build on the manuscript as a reliable and valuable source of information.

Response: We thank Reviewer 2 for their thoughtful summary of our work and assessment of its significance to the study of the protein Echinoid as well as to the fields of growth regulation, cell competition, and Hippo signalling. We are glad they find our data to be of high quality. We have taken every effort to address all of the points made by the Reviewer.

SUGGESTIONS TO AUTHORS

The authors attempted to determine whether Echinoid influences clone growth via a cell adhesion function or a signalling function, using a transgenic rescue approach, although the results were not completely conclusive. Tailored Echinoid CRISPR alleles are probably the best way to approach this, but this is not trivial. Instead, perhaps the authors could use an alternate wing disc Gal4 driver that does not have a phenotype on its own (like nubbin-Gal4 did). I agree with the authors that Echinoid probably influences clone growth via a cell adhesion function, especially given that Echinoid and E-cad LOF clones phenotypically resemble each other, but it could be tested more rigorously.

Response: We appreciated the suggestion to test this hypothesis more rigorously. After conducting more experiments, we found this issue to be quite complicated. First, we repeated our *ed*-RNAi knockdown using the wing pouch driver *rn-Gal4* as a potential replacement for *nub-Gal4*. Unfortunately, we found that both Gal4 driver lines have smaller wing sizes than *Oregon-R*. That there is variability in wing size even between “control” lines highlights that it is essential to properly control for genetic background when interpreting our results.

We also realized that the focusing on genetic rescue in our initial set of experiments was an unnecessary complex starting point for dissecting the roles of different Ed domains in wing growth and morphogenesis, and that we could learn more through simple overexpression experiments. We verified with multiple drivers that overexpression of full-length Ed reduces wing size. Overexpression of two different GFP-tagged alleles with C-terminal truncations (*UAS-ed^{ΔC}-GFP* [Laplante and Nilson, 2011] and *UAS-ed^{ΔIntra}-GFP* [gift of J-C. Hsu]) gave different results: *UAS-ed^{ΔC}-GFP* overexpression reduced wing size and circularity, whereas *UAS-ed^{ΔIntra}-GFP* had no effect on wing size or shape. That expression of at least one *ed* transgene lacking the intracellular domain led to reductions in wing size and roundness, as well as rescue of overgrowth and roundness in *ed^{slH8}/ed^{IF20}*, suggests that the C terminal domain is not essential.

With these new data, we have substantially revised and rewritten the section titled “Echinoid regulates the size and shape of the adult wing” as well as the associated Figure (Fig. 7) and Supplementary Figure (Fig. S6).

To make the manuscript easier to read, the authors could streamline and shorten the Results and Discussion sections.

Response: We appreciate this suggestion. We have edited our paper for brevity and clarity throughout the revision process.

Reviewer 3

SUMMARY OF THE ADVANCE MADE IN THIS PAPER AND ITS POTENTIAL SIGNIFICANCE TO THE FIELD

This manuscript explores the functions of Echinoid (Ed) and other cell adhesion molecules in organ growth. Ed has been implicated in the regulation of several signaling pathways, including EGFR, Notch, and Hippo, and appears to be involved in complex molecular interactions that may relate to endomembrane trafficking, cell adhesion, and possibly signaling. Careful genetic experiments are presented that strive to explain paradoxical outcomes of ed loss: small clones of cells that are mutant for ed are eliminated via apoptosis, whereas large areas of ed mutant tissue exhibit overgrowth. The first outcome is attributed to a reduction in Diap1 and a

concomitant increase in apoptosis in small ed clones.

Surprisingly, surrounding areas show increased Diap1, creating a possibility for cell competition. The second outcome may stem from an extension of the larval stage due to hyperactivation of Ilp8 and possibly other unknown mechanisms that contribute to a failure to arrest growth.

Some of the reported results are in direct contradiction with a previously published study (Yue et al., 2012). While some of these contradictions can be simply explained by the other study using a cell lethal mutation that favored the overgrowth of ed mutant tissue, others cannot be explained in a simple way, such as the discrepancy in Diap1 expression and Yki nuclear localization in ed mutant cells. Overall, this manuscript proposes interesting, if not provocative, new models for explaining the complex functions of Ed in development and is written very well. With rigorous data and thoughtful analysis, it will contribute to a continued discourse in the field by highlighting the complexities surrounding Ed functions.

Response: We appreciate Reviewer 3's clear overview of our work and are glad that they find our work to be rigorous, thoughtful, and well-written. We appreciate the constructive feedback they provided and have done our best to address all of their points.

At present, it is difficult to evaluate the significance of the involvement of Diap1 in the survival of ed mutant cells shown in Fig. 4G-H', for two reasons. First, the difference between H and G is not as striking as some of the other results; second, since ed clone survival appears to be rather sensitive to the duration and timing of heat shock (Fig. 3), the result shown in H might be attributed to a fluctuation in experimental conditions. A more solid demonstration of Diap1 involvement in ed-mediated apoptosis may be obtained e.g. via quantification of the differences in clone number or size. Another possibility is to repeat this experiment using low clone density, as the differences may become more apparent (like the comparison in Fig. 4D vs. 4C).

Response: We are grateful to the Reviewer for this feedback. As suggested, we repeated the experiment using a shorter heat shock to generate clones at lower density, which make the extent of rescue much more apparent (Thank you!). We have updated the figure to include these data which better illustrate the rescue phenotype.

There is a discrepancy between nuclear Yki localization reported in ed mutant cells in Yue et al. vs. no changes observed in this manuscript. Yue et al. used antibody staining for Yki, whereas this manuscript used a well-validated CRISPR-generated line where Yki is tagged with GFP. I suggest checking Yki localization using antibody staining in addition to using this line, to confirm this result. Also, since Yki target genes show higher expression in cells surrounding the clones, it is puzzling why Yki levels would not show an increase in those cells. The Fat explanation would still require increased Yki nuclear localization as a downstream outcome, so higher nuclear Yki would be expected in surrounding cells. It may also be possible to explain the induction of Yki target genes in surrounding cells through decreased Hpo signaling due to mechanical effects stemming from actomyosin cable formation in those cells. But this latter explanation would still expect an increase in nuclear Yki.

Response: We appreciated the Reviewer's suggestion and repeated the experiment, this time including an anti-Yki antibody stain in addition to the anti-GFP antibody which would recognize the transgenic Yki:GFP only. We got the same result as before: no noticeable difference in Yki nuclear localization within, next to, or outside of ed clones. We have updated Figure 5E to show the result from our replication experiment, with the Yki antibody stain now included.

We have also added additional comments regarding the Yki localization data presented in Yue et al. (2012):

“[Our] result is in contrast to data presented by Yue et al. (2012), which showed strong nuclear localization of Yki in ed cells and diffuse staining in wild-type cells. We found their result surprising, given that Yki antibody stains typically produce a “honeycomb-like” pattern in wild-type cells in the disc proper of the wing imaginal disc due to nuclear exclusion; nuclear relocalization of Yki caused by Hippo pathway mutations generally results in more uniform staining of the cytoplasm and nucleus rather than in strong, distinctly-nuclear staining (see, for example, Figure 3C-D in Dong et al., 2007)”

Fig. 2J,K should indicate in the figure itself and/or in the legend which cell lethal mutation was used (gene and allele).

Response: We thank the Reviewer for catching this oversight. We updated the figure legend to say, “The tester chromosome carries a recessive cell lethal allele l(2)cl-L31 resulting in the absence of wild-type twin spots when homozygous.”

Check sentence starting at line 381 (“We also reduced...”), looks like “to” is missing before “drive”.

Response: We thank the Reviewer for catching this typo, which we have now fixed.

In Fig. 6B’, apoptosis is observed throughout the area where ed is knocked down. How can this result be reconciled with Fig. 4F’, where apoptosis was mostly observed at the edges of the clones, and with a model where apoptosis is driven by the difference in Diap1 levels in adjacent cells?

Response: We appreciate this question from the Reviewer. We did not wish to suggest that border competition is the sole driver of apoptosis in *ed* cells so we have clarified the language in our manuscript. To emphasize this point, we added this additional statement in the Discussion: “This elevated baseline propensity for apoptosis in mutant cells, enhanced at clonal boundaries with wild-type cells, resembles the phenotype resulting from *Minute* mutations (Akai et al., 2021; Baumgartner et al., 2021; Coelho et al., 2005; Recasens-Alvarez et al., 2021)”

We thank the reviewers again for their thoughtful and constructive feedback, which has greatly improved the quality of our revised manuscript.

Second decision letter

MS ID#: dev.204572R1

MS Title: The cell adhesion molecule Echinoid promotes tissue survival and separately restricts tissue overgrowth

Authors: Danielle Spitzer; William Sun; Anthony Rodríguez-Vargas; Iswar Hariharan
Article Type: Research Article

Dear Dr Hariharan,

I am happy to tell you that your manuscript has been accepted for publication in Development, pending our standard publication integrity checks.

Reviewer 1

The authors have addressed my comments and I am happy to recommend publication in Development.

Reviewer 2

The manuscript has been improved and acceptance for publication is recommended.